# Enhancing NSCLC recurrence prediction with PET/CT habitat imaging, ctDNA, and integrative radiogenomics-blood insights

Sheeba J. Sujit [1,18], Muhammad Aminu [1,18], Tatiana V. Karpinets[2], Pingjun Chen [1], Maliazurina B. Saad[1], Morteza Salehjahromi[1], John D. Boom [1,3], Mohamed Qayati[1], James M. George [1,4], Haley Allen[5], Mara B. Antonoff [6], Lingzhi Hong [1,7], Xin Hu[7], Simon Heeke [7], Hai T. Tran[7], Xiuning Le [7], Yasir Y. Elamin [7], Mehmet Altan [7], Natalie I. Vokes [2,7], Ajay Sheshadri [8], Julie Lin[8], Jianhua Zhang [2], Yang Lu [9], Carmen Behrens[7], Myrna C. B. Godoy[10], Carol C. Wu [10], Joe Y. Chang[11], Caroline Chung[11,12], David A. Jaffray[1,12,13], Ignacio I. Wistuba[14], J. Jack Lee [15], Ara A. Vaporciyan[6], Don L. Gibbons [7], John Heymach [7], Jianjun Zhang [2,7,16,17,19], Tina Cascone [7,19] & Jia Wu [1,7,12,19] ✉

While we recognize the prognostic importance of clinicopathological measures and circulating tumor DNA (ctDNA), the independent contribution of quantitative image markers to prognosis in non-small cell lung cancer (NSCLC) remains underexplored. In our multi-institutional study of 394 NSCLC patients, we utilize pre-treatment computed tomography (CT) and [18]F-fluorodeoxyglucose positron emission tomography (FDG-PET) to establish a habitat imaging framework for assessing regional heterogeneity within individual tumors. This framework identifies three PET/CT subtypes, which maintain prognostic value after adjusting for clinicopathologic risk factors including tumor volume. Additionally, these subtypes complement ctDNA in predicting disease recurrence. Radiogenomics analysis unveil the molecular underpinnings of these imaging subtypes, highlighting downregulation in interferon alpha and gamma pathways in the high-risk subtype. In summary, our study demonstrates that these habitat imaging subtypes effectively stratify NSCLC patients based on their risk levels for disease recurrence after initial curative surgery or radiotherapy, providing valuable insights for personalized treatment approaches.

Results for treating non-small cell lung cancer (NSCLC) remain dismal despite advancements in surgical methods, radiation treatments, and the use of immune checkpoint inhibitors. With early-stage or locally advanced NSCLC, between 30% and 55% of patients will experience disease recurrence and eventually die. Epidermal growth factor tyrosine kinase inhibitors and immunotherapy with chemotherapy have been approved for use in adjuvant and neoadjuvant settings, respectively[1–3]. However, non-invasive markers that can predict disease relapse are urgently needed to stratify patients and direct treatment escalation or de-escalation in the perioperative setting.

Computed tomography (CT) and [18]F-fluorodeoxyglucose positron emission tomography ([18]F-FDG PET) are non-invasive tools routinely

used for initial staging, radiation treatment planning, and response evaluation for patients with NSCLC. Recently, radiomic features have shown prognostic ability with regard to NSCLC outcomes[4–10]. However, considering that CT and PET are distinct (anatomic vs. metabolic) and complementary, these studies predominantly captured tumor characteristics using a single modality, representing a partial view of the tumor. Another fundamental limitation is the lack of profiling of regional variation within the gross tumor[11]. Although texture analysis has been applied to measure intratumoral heterogeneity, it assumes a homogeneous mixture within the tumor that has been shown untrue at the tissue and cellular levels[12].

According to multiregional molecular profiling conducted by our team and others[13–15], the tumor is a complex ecosystem with regional variation. Intratumor heterogeneity is a dynamic factor that promotes tumor growth and resistance, and typically indicates poor clinical prognosis[16,17]. Though the molecular heterogeneity of tumor cells has been characterized from the view of cancer biology, it is unclear how this heterogeneity would appear on macroscopic radiography scans[17]. Given that imaging depicts spatial heterogeneity in the architecture of individual tumors, the emerging tool of habitat imaging is suggested to capture these regional distinctions, with clinical implications in different cancer types[18,19].

Habitat Imaging is a modern approach used in cancer imaging to identify tumor subregions or 'habitats' that share imaging traits that are characterized by imaging biomarkers[20]. We aimed to use habitat imaging patterns to define intrinsic radiological subtypes as manifested on CT and FDG-PET, and further test their clinical relevance by stratifying patients' risk of recurrence after curative surgery or radiotherapy using multi-institutional, multi-modality (imaging and genomics) cohorts of resected NSCLC. We also demonstrated the added value of circulating tumor DNA (ctDNA), an emerging blood marker, in conjunction with our proposed habitat imaging subtypes. Finally, radiogenomic analysis was done to determine the biology of these habitat imaging subtypes.

## Results

### Patient characteristics of multicenter data

We retrospectively collected multimodal data (CT and PET, gene expression, clinical characteristics) from 4 independent NSCLC cohorts totaling 394 patients (Fig. 1). The discovery ($n = 199$) and validation ($n = 195$) cohorts showed similar demographic distributions (Table 1). Of the total 394 NSCLC patients included in the study, 224 (57%) were men, and the median age was 67 years. Almost 42% of patients had stage I NSCLC. The median recurrence time was 43

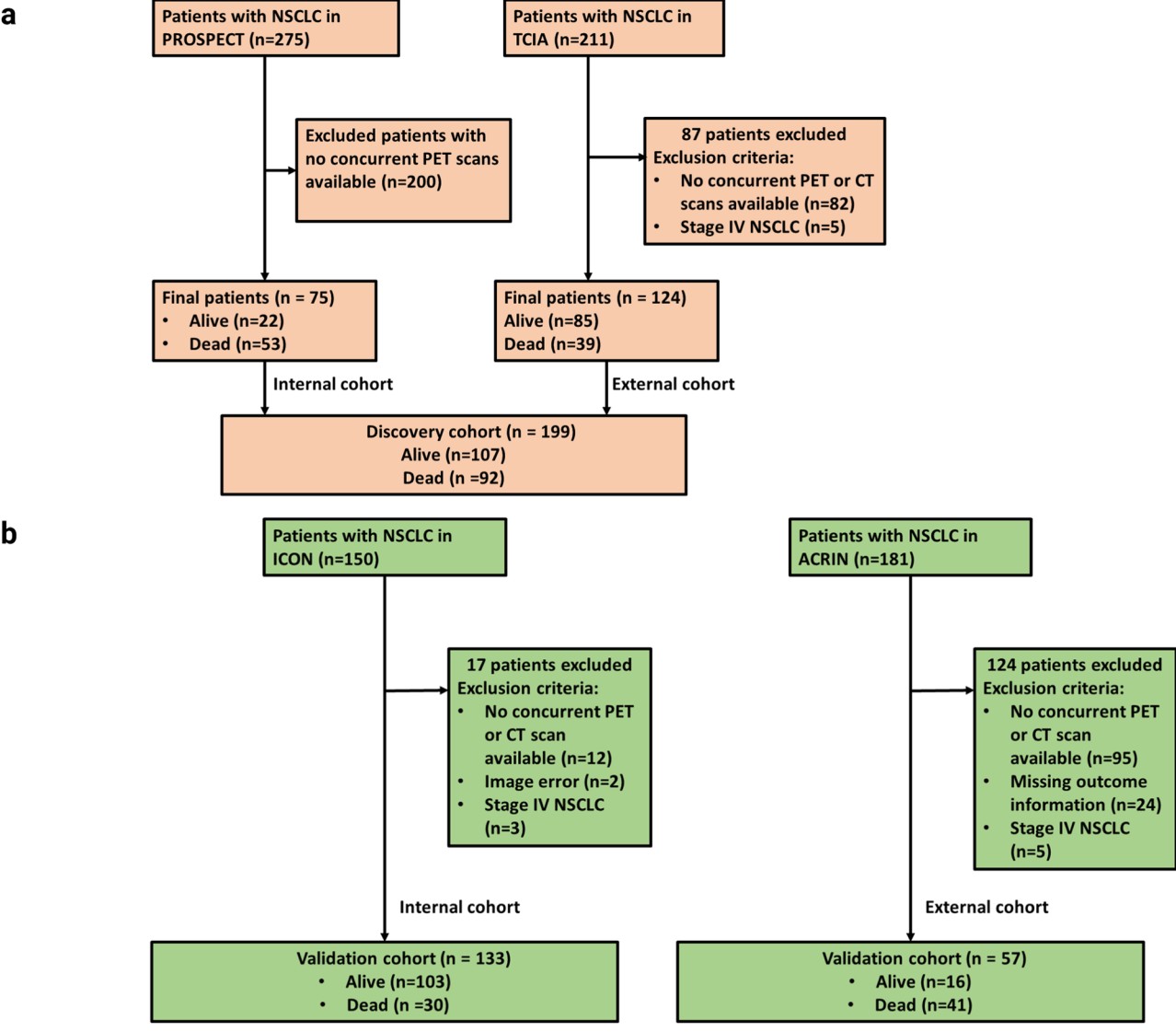

Fig. 1 | Study cohorts and their characteristics. a The discovery cohort included 75 patients from the PROSPECT database and 124 patients from the TCIA database. b The validation cohorts consisted of 133 patients from the ICON database and 62 patients from ACRIN database.

**Table 1 | Summary of demographic and clinical characteristics from the study cohorts**

| Parameter | Discovery cohort (n = 199) | Internal validation cohort (n = 133) | External validation cohort (n = 62) | P - value |
|---|---|---|---|---|
| **Age** | | | | |
| **Median (SD)** | 68.0 (10.6) | 67.0 (9.5) | 66.5 (9.25) | 0.0341[i] |
| **Gender**, n (%) | | | | 0.5753[i] |
| Male | 122 (61%) | 63 (47%) | 39 (63%) | |
| Female | 77 (39%) | 70 (53%) | 23 (37%) | |
| **P Stage (AJCC 7th ed.)**, n (%) | | | | 0.4751[i] |
| 0 | 4 (1.96%) | 1 (0.73%) | - | |
| I | 114 (55.88%) | 50 (36.76%) | - | |
| II | 41 (20.09%) | 47 (35.33%) | 2 (3%) | |
| III | 40 (19.60%) | 34 (25%) | 60 (97%) | |
| **Histology**, n (%) | | | | 0.0375 |
| Adenocarcinoma | 154 (80%) | 89 (67%) | - | |
| Squamous cell | 39 (20%) | 31 (23%) | - | |
| Other | 6 (3%) | 13 (10%) | - | |
| **Smoking History**, n (%) | | | | 0.1056 |
| Smoking | 157 (79%) | 111 (83%) | - | |
| Never | 42 (21%) | 22 (17%) | - | |
| **RNA Sequence**, n (%) | | | | |
| Available | 115 (58%) | 93 (70%) | - | |
| **ctDNA**, n (%) | | | - | |
| At baseline prior to surgery | - | 72 (54%) | - | |
| Clearance status after surgery | - | 50 938%) | - | |
| **Median Follow-up time (months)** | 50 | 44 | 20 | 0.07[i] |
| **Median OS time (months)** | 35 | 20 | 13 | 0.24[i] |
| **Recurrence Free Survival**, n (%) | | | | 0.5658 |
| Recurrence (1) | 73 (37%) | 42 (32%) | - | |
| No Recurrence (0) | 126 (63%) | 91 (68%) | - | |
| **Overall Survival**, n (%) | | | | 0.03751[i] |
| Dead (1) | 92 (46%) | 30 (23%) | 41 (72%) | |
| Alive (0) | 107 (54%) | 103 (77%) | 16 (28%) | |

[i] P – value was calculated using Pearson's Chi-square test (two-sided) comparing Discovery cohort and Integrated validation cohorts.

months for the discovery cohort and 35 months for the internal validation cohort ($P = 0.56$). Average treatment period for ICON cohort is 37 days, and average treatment period for PROSPECT cohort is 36.5 days. The overall study design of the habitat imaging–based radiomics analysis is shown in Fig. 2.

### Identification of habitat subregions with distinct imaging phenotypes

Eight tumor subregions (clusters) emerged, and we confirmed their consistency across discovery and validation cohorts, as embedded in UMAP (Fig. 3a, b). The annotation of individual subregions by mapping onto the superpixel-level radiomic space revealed their distinct radiographical patterns consistently across different cohorts (Supplementary Fig. 1a). Each of these eight habitat subregions had a distinct imaging phenotype, which was shown in Fig. 3c, Supplementary

Fig. 1b and Supplementary Data 7. For clusters 3 and 4, CT Hounsfield Units (HU) number is high, whereas PET SUV and PET entropy values are low (L). Based on the violin plots in Supp. Fig. 1b, we observed that the CT Hounsfield Units (HU) numbers or the distribution of the CT HU numbers in cluster 3 and 4 is much higher than the distribution of PET SUV values in the same clusters. This indicates a denser tissue. For clusters 7 and 8, CT number and PET SUV are high and CT entropy is low, respectively. Habitat maps of six cases of patients from the discovery cohort are shown in Fig. 3d, and they reveal a significant prevalence of clusters 7 and 8 in high-risk patients and clusters 3 and 4 in low-risk patients.

### Imaging subtypes in the discovery and validation cohorts

From the eight clusters identified upon integrating the CT and PET scans from the tumor region of each patient, ninety-two multiregional spatial interaction (MSI) features were extracted to measure intratumoral spatial heterogeneity (Table 2, Fig. 2b). We identified three imaging subtypes after consensus clustering these imaging features (Supplementary Fig. 3a), which was the optimal solution for minimizing incremental change in the area under the CDF curve while maximizing consensus within subtypes This was further confirmed by consensus matrix heatmaps with k = 3, which optimally represented the data pattern of patients in the discovery and validation cohorts. When we visualized the matrix heatmaps and the proportion increase of area under the CDF curve, k = 3 was the largest number of clusters considered (Supplementary Fig. 3b, Supplementary Data 9). In the discovery cohort ($n = 199$), the subtypes based on MSI features were prognostic of RFS, with $P = 1\text{e-}05$ while stratifying patients into low-risk, high-risk, and intermediate-risk subtypes (Fig. 4a). These imaging subtypes remained prognostic of RFS in the internal validation cohort ($n = 133$), with $P = 0.024$ (Fig. 4b). These subtypes were also prognostic of OS in the discovery, internal (Fig. 4c, d) and external validation cohorts (Fig. 4e).

Imaging subtypes were compared with classical radiomic approach. Radiomic features were extracted from the entire tumor ROI using the pyradiomics python package[21]. Classical radiomics model did not achieve robust stratification of patients in predicting RFS and OS during validation (Supplementary Fig. 4, Supplementary Data 10), where our habitat imaging subtypes outperformed the classical radiomics approach with significant net reclassification improvement (Supplementary Table 1, Supplementary Data 1).

By mapping the habitat imaging subtypes to the space of the original MSI features and conventional CT and PET features across the discovery and validation cohorts (Supplementary Fig. 5, Supplementary Data 11), we observed that the high-risk population tended to have escalated volume of cluster 7 and abundant clusters 4 and 7 on the tumor border. The pairwise comparison of key habitat features between different risk groups is presented in Supplementary Fig. 6 (Supplementary Data 12).

### Independent prognostic value of habitat imaging subtypes beyond conventional risk predictors

When compared with known clinicopathological risk factors, the imaging subtypes were top-ranked predictors of RFS (Table 3). In multivariate analysis, imaging subtypes remained prognostic of RFS after adjustment for clinicopathological factors, including age, sex, smoking, and conventional imaging metrics (CT: tumor volume; PET: $SUV_{max}$, metabolic tumor volume, total lesion glycolysis) in the integrated cohort. The imaging subtypes were similarly prognostic of OS (Table 4). Univariate and multivariate analysis, after adjustment for tumor volume, reveals the independent impact of habitat imaging subtypes in both the discovery and validation cohorts (Supplementary Table 2, Supplementary Table 3, Supplementary Data 2).

We observed 58%, 59%, and 64% correlation between three habitat subtypes and the stratification by MTV, TLG, or tumor volume,

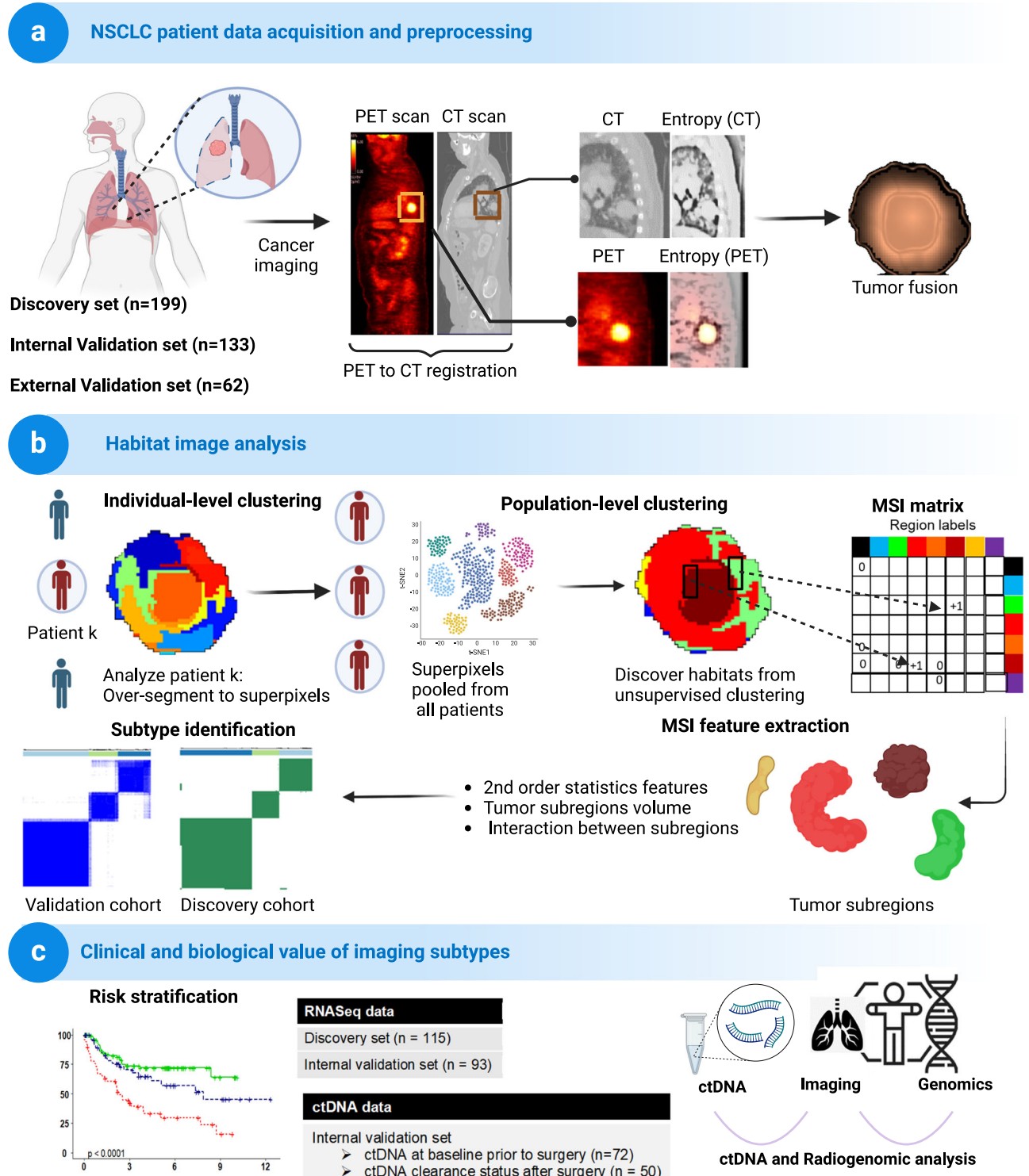

**Fig. 2 | Study design of habitat imaging–based radiomic analysis of NSCLC patients. a** Overview of the data collected and pre-processing stages of the CT and [18]F-FDG PET images. The pre-processing involved [18]F-FDG PET to CT registration and fusion of segmented tumor regions from [18]F-FDG PET and CT images along with their local entropy maps. **b** The habitat imaging analysis framework consisted of a 2-stage clustering process: Individual-level clustering, where tumor regions of each patient were over-segmented into superpixels; and population-level clustering, where clustering was performed on superpixels pooled from all patients to discover distinct tumor subregions. The multiregional spatial interaction (MSI) matrix summarizes the co-occurrence statistics among different tumor subregions. The 92 MSI features extracted from the MSI matrix identified patient subtypes. **c** Radiogenomic analysis along with ctDNA metrics confirm the clinical and biological relevance of the identified imaging subtypes. Created with BioRender.com.

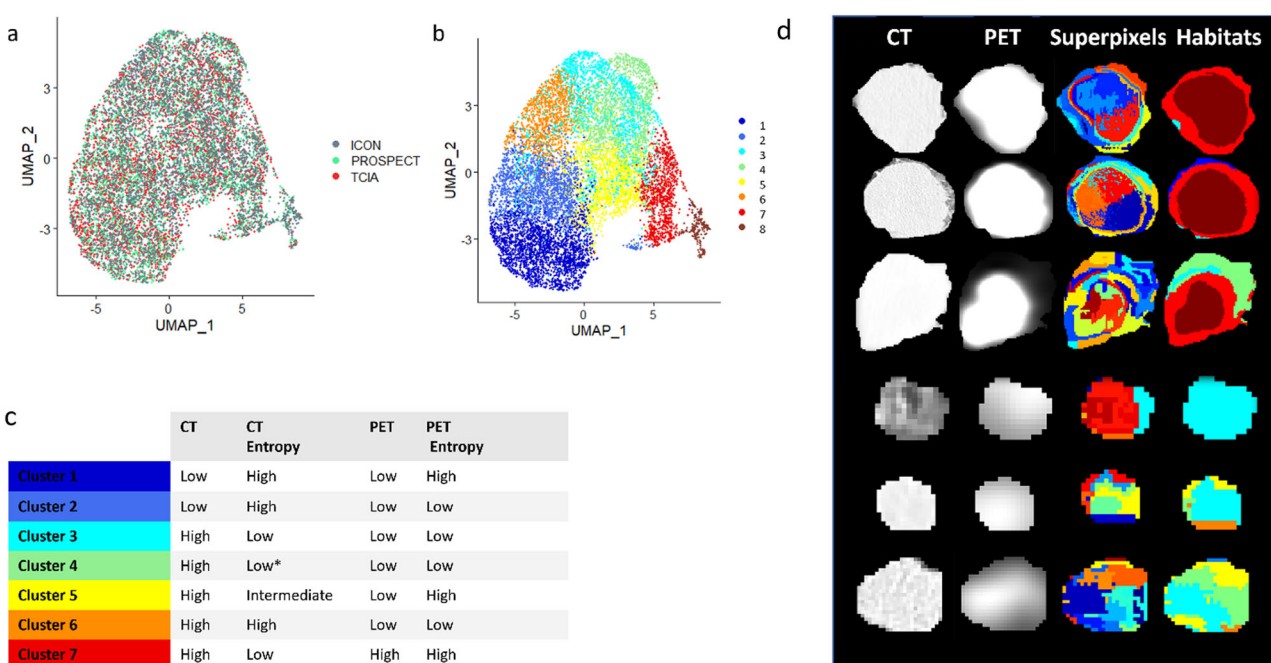

**Fig. 3 | Systematic benchmarks of unsupervised clustering analysis of tumor region.** Visualization in UMAP: **a** distribution of superpixels across the discovery (PROSPECT, TCIA) and validation ICON cohorts. **b** the 8 clusters identified using Louvain clustering by dimension reduction. **c** Imaging interpretation of the eight cluster regions using High, Low, Intermediate levels. **d** Examples of habitat maps of patients with disease recurrence and no recurrence from the discovery set. Rows 1, 2 and 3 show habitat maps of patients who has recurrence of disease after 5, 47 and 37.5 months, respectively. They show high volume of clusters 7 and 8. Rows 4, 5 and 6 show habitat maps of patients who has no recurrence of disease showing high volume of clusters 3 and 4.

**Table 2 | Ninety-two quantitative imaging features extracted from the multiregional spatial interaction (MSI) matrix to measure intratumoral spatial heterogeneity on PET/CT Habitat Map**

| Feature name | Feature description |
|---|---|
| MSI 1 – MSI 4 | 2nd-order statistical features. |
| MSI 5 – MSI 12 | absolute tumor subregions volume (SR1 – SR8). |
| MSI 13 – MSI 20 | interaction (absolute) between tumor subregions and border. |
| MSI 21 – MSI 27 | interaction (absolute) between SR1 and the remaining subregions, i.e., MSI 21 = SR1 ∩ SR2, MSI 22 = SR1 ∩ SR3, ..., MSI 27 = SR1 ∩ SR8. |
| MSI 28 – MSI 33 | interaction (absolute) between SR2 and SR3, SR4, SR5, S6, S7 and SR8, i.e., MSI 28 = SR2 ∩ SR3, MSI 29 = SR2 ∩ SR4, MSI 30 = SR2 ∩ SR5, MSI 31 = SR2 ∩ SR6, MSI 32 = SR2 ∩ SR7, MSI 33 = SR2 ∩ SR8. |
| MSI 34 – MSI 38 | interaction (absolute) between SR3 and SR4, SR5, SR6, SR7 and SR8 i.e., MSI 34 = SR3 ∩ SR4, MSI 35 = SR3 ∩ SR5, MSI 36 = SR3 ∩ SR6, MSI 37 = SR3 ∩ SR7, MSI 38 = SR3 ∩ SR8. |
| MSI 39 – MSI 42 | interaction (absolute) between SR4 and SR5, SR6, SR7 and SR8 i.e., MSI 39 = SR4 ∩ SR5, MSI 40 = SR4 ∩ SR6, MSI 41 = SR4 ∩ SR7, MSI 42 = SR4 ∩ SR8. |
| MSI 43 – MSI 45 | interaction (absolute) between SR5 and SR6, SR7 and SR8 i.e., MSI 43 = SR5 ∩ SR6, MSI 44 = SR5 ∩ SR7, MSI 45 = SR5 ∩ SR8. |
| MSI 46 – MSI 47 | interaction (absolute) between SR6 and SR7 and SR8 i.e., MSI 46 = SR6 ∩ SR7, MSI 47 = SR6 ∩ SR8. |
| MSI 48 | interaction (absolute) between SR7 and SR8, i.e., MSI 48 = SR7 ∩ SR8. |
| MSI 49 – MSI 56 | percentage of tumor subregions volume (SR1 – SR8). |
| MSI 57 – MSI 64 | normalized interaction (percentage) between tumor subregions and border. |
| MSI 65 – MSI 71 | normalized interaction (percentage) between SR1 and the remaining subregions, i.e., MSI 65 = SR1 ∩ SR2, MSI 66 = SR1 ∩ SR3, ..., MSI 71 = SR1 ∩ SR8. |
| MSI 72 – MSI 77 | normalized interaction (percentage) between SR2 and SR3, SR4, SR5, SR6, SR7 and SR8 i.e., MSI 72 = SR2 ∩ SR3, MSI 73 = SR2 ∩ SR4, MSI 74 = SR2 ∩ SR5, MSI 75 = SR2 ∩ SR6, MSI 76 = SR2 ∩ SR7, MSI 77 = SR2 ∩ SR8. |
| MSI 78 – MSI 82 | normalized interaction (percentage) between SR3 and SR4, SR5, SR6, SR7 and SR8 i.e., MSI 78 = SR3 ∩ SR4, MSI 79 = SR3 ∩ SR5, MSI 80 = SR3 ∩ SR6, MSI 81 = SR3 ∩ SR7, MSI 82 = SR3 ∩ SR8. |
| MSI 83 – MSI 86 | normalized interaction (percentage) between SR4 and SR5, SR6, SR7 and SR8 i.e., MSI 83 = SR4 ∩ SR5, MSI 84 = SR4 ∩ SR6, MSI 85 = SR4 ∩ SR7, MSI 86 = SR4 ∩ SR8. |
| MSI 87 – MSI 89 | normalized interaction (percentage) between SR5 and SR6, SR7 and SR8 i.e., MSI 87 = SR5 ∩ SR6, MSI 88 = SR5 ∩ SR7, MSI 89 = SR5 ∩ SR8. |
| MSI 90 – MSI 91 | normalized interaction (percentage) between SR6 and SR7 and SR8 i.e., MSI 90 = SR6 ∩ SR7, MSI 91 = SR6 ∩ SR8. |
| MSI 92 | normalized interaction (percentage) between SR7 and SR8, i.e., MSI 92 = SR7 ∩ SR8. |

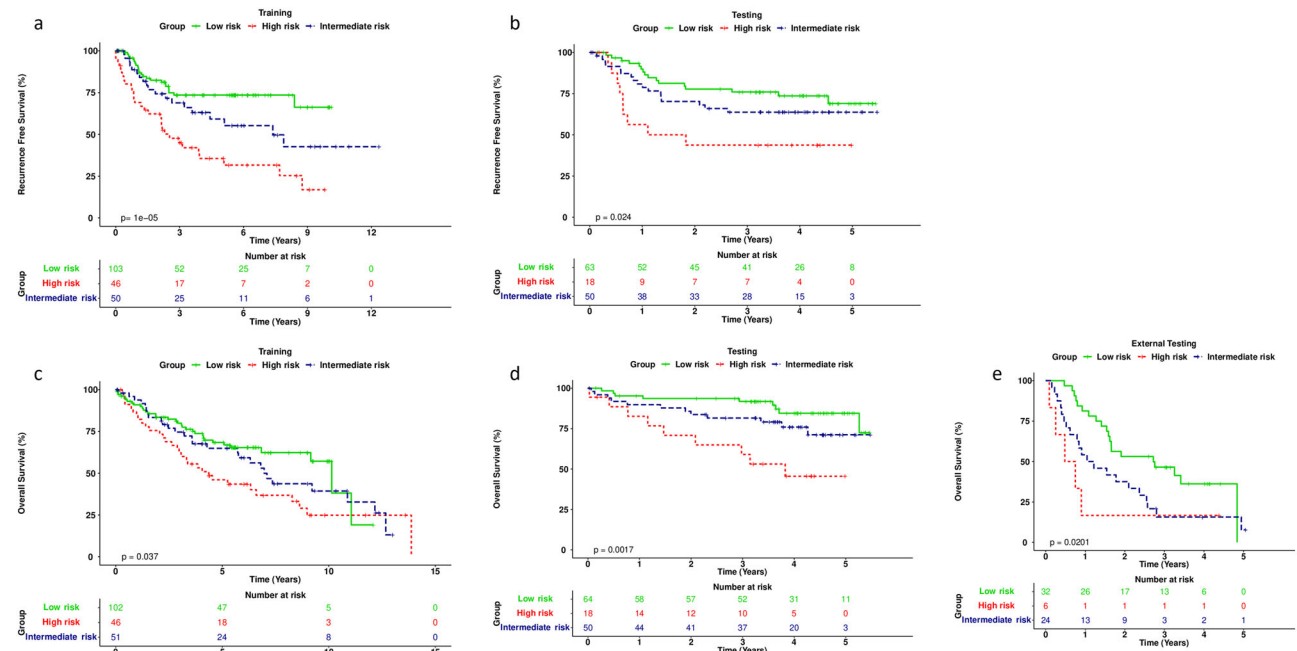

**Fig. 4 | Imaging subtypes were prognostic for recurrence-free survival in NSCLC patients.** Kaplan-Meier curve comparing RFS of individuals with low risk (green), high risk (red), and intermediate risk (purple) subtypes with $P = 1e-05$ by log-rank test in the discovery cohort (**a**) and $P = 0.024$ by log-rank test in the validation cohort (**b**). Kaplan-Meier curve comparing OS of individuals with low risk (green), high risk (red), and intermediate risk (purple) subtypes with $P = 0.037$ by log-rank test in the training cohort (**c**) $P = 0.0017$ by log-rank test in the internal validation cohort (**d**) and $P = 0.0201$ by log-rank test in the external validation cohort (**e**). Source data are provided as a Source Data file.

**Table 3 | Univariate and multivariate Cox regression analysis of recurrence-free survival (RFS) in the integrated study cohort**

| Variables | Univariate | | Multivariate | |
|---|---|---|---|---|
| | **P value** | **HR (95% CI)** | **P value** | **HR (95% CI)** |
| **Imaging Subtypes** | | | | |
| High – Risk | Reference | | | |
| Intermediate - Risk | 0.002** | 0.48 (0.31–0.76) | 0.015* | 0.52 (0.31–0.88) |
| Low - Risk | 1.88e-07*** | 0.31 (0.20–0.48) | 0.001** | 0.38 (0.21–0.68) |
| Age | 0.889 | 1 (0.98–1) | --- | --- |
| Male vs Female | 0.526 | 1.1 (0.78–1.6) | --- | --- |
| Non-Smoker vs Smoker | 0.890 | 0.97 (0.62–1.5) | --- | --- |
| Histologic type[a] | 0.075 | 0.69 (0.46–1) | 0.203 | 0.75 (0.48–1.17) |
| Treatment type[b] | 3.71e-07 *** | 2.6 (1.8–3.9) | 0.0005*** | 2.07 (1.37–3.13) |
| Metabolic tumor volume (MTV) | 0.000216 *** | 1.5 (1.2–1.8) | 0.801 | 0.90 (0.39–2.09) |
| Total Lesion Glycolysis (TLG) | 0.001** | 1 (1–1) | 0.657 | 1.01 (0.96–1.06) |
| SUV (max) | 0.074 | 1 (1–1) | 0.396 | 0.99 (0.97–1.01) |
| SUV (mean) | 0.411 | 1 (0.97–1.1) | --- | --- |
| Tumor volume | 0.000161 *** | 1.2 (1.1–1.4) | 0.973 | 0.99 (0.70–1.41) |

[a]Adenocarcinoma vs other types (reference).
[b]Chemotherapy vs other types (reference).
With the high-risk subtype as the baseline, the low-risk and intermediate-risk imaging subtypes remained independent predictors of RFS (hazard ratio [HR] = 0.38, $P = 0.001$**; HR = 0.52, $P = 0.015$*, respectively), and the low-risk imaging subtype remained an independent predictor of OS (HR = 0.52, $P = 0.025$*) in multivariate analysis. * $P < 0.05$; ** $P < 0.01$; *** $P < 0.001$. All statistical tests were two-sided, with $P < 0.05$ indicative of a statistically significant difference. Source data are provided as a Source Data file.

respectively (Supplementary Table 4, Supplementary Data 3), indicating habitat analysis learns radiological patterns beyond these conventional CT and PET metrics. Tumor volume was associated with baseline ctDNA status, while marginal statistical difference was observed between tumor volume and serial ctDNA clearance status (Supplementary Fig. 2, Supplementary Data 8). When analyzing the primary tumor location, we observed that only the right upper lobe is prognostic in our cohorts that is independent of habitat imaging. High-

risk and intermediate-risk imaging subtypes remained prognostic of RFS and OS in the multivariate analysis after adjustment for tumor and lobe location information (Supplementary Tables 5 and 6, Supplementary Data 4). There is no correlation between radiotherapy-related parameters (post-operative radiotherapy and radiation dose) and survival after adjusting them in univariate and multivariate analyzes (Supplementary Tables 7 and 8, Supplementary Data 5). The oncogene mutation status (i.e., EGFR mutation and ALK fusion) exhibits no

**Table 4 | Univariate and multivariate Cox regression analysis of overall survival (OS) in the integrated study cohort**

| Variables | Univariate | | Multivariate | |
|---|---|---|---|---|
| | *P* value | HR (95% CI) | *P* value | HR (95% CI) |
| **Imaging Subtypes** | | | | |
| High – Risk | Reference | | | |
| Intermediate - Risk | 0.015* | 0.58 (0.37–0.90) | 0.136 | 0.68 (0.41–1.13) |
| Low - Risk | 0.000123 *** | 0.43 (0.27–0.66) | 0.025* | 0.52 (0.29–0.92) |
| Age | 0.009* | 1 (1–1) | --- | --- |
| Male vs Female | 0.005** | 1.7 (1.2–2.5) | --- | --- |
| Non-Smoker vs Smoker | 0.104 | 1.5 (0.92–2.5) | --- | --- |
| Histologic type ᵃ | 0.000693 *** | 0.52 (0.36–0.76) | 0.005** | 0.55 (0.36–0.83) |
| Treatment type ᵇ | 0.023* | 1.6 (1.1–2.3) | 0.33 | 1.24 (0.81–1.90) |
| Metabolic tumor volume (MTV) | 0.001** | 1.4 (1.1–1.7) | 0.175 | 1.88 (0.75–4.75) |
| Total Lesion Glycolysis (TLG) | 0.019* | 1 (1–1) | 0.476 | 0.97 (0.91–1.05) |
| SUV (max) | 0.116 | 1 (1–1) | 0.364 | 0.99 (0.96–1.01) |
| SUV (mean) | 0.092 | 1 (0.99–1.1) | --- | --- |
| Tumor volume | 0.004** | 1.2 (1.1–1.3) | 0.262 | 0.81 (0.56–1.17) |

ᵃ Adenocarcinoma vs other types (reference).

ᵇ Chemotherapy vs other types (reference).

* *P* < 0.05; ** *P* < 0.01; *** *P* < 0.001. All statistical tests were two-sided, with *P* < 0.05 indicative of a statistically significant difference. Source data are provided as a Source Data file.

correlation with patient survival, as illustrated in the table (Supplementary Tables 9 and 10, Supplementary Data 6), which may be due to the limited sample size of patients with tumors harboring these genomic alterations.

We then carried out a detailed subgroup analysis as stratified by clinicopathological features (including stage, gender, smoking status, age, histology, and treatment) and tumor volume (Supplementary Data 13), and found that habitat imaging subtypes can further stratify patient outcomes within individual subgroups in RFS (Supplementary Fig. 7) and OS prediction (Supplementary Fig. 8). Also, 3 patients with stage II adenocarcinoma could be stratified into different imaging subtypes with distinct risk levels of cancer recurrence based on their pre-surgery PET/CT as shown in Fig. 5. Since a considerable proportion of early-stage NSCLC recurs after surgery, this type of risk stratification may enable the selection of high-risk patients for peri-operative treatment.

**Imaging subtype complements ctDNA from blood in predicting survival**

We observed that the high- and intermediate-risk subtypes based on habitat imaging had a higher percentage of patients with ctDNA detected in blood before surgery compared to the low-risk subtype (Fig. 6a). The presence of persistent ctDNA was observed in 67% of high-risk patients, and by contrast, 62% of low-risk patients did not have ctDNA detected (Fig. 6b). Interestingly, 45% of intermediate risk patients had ctDNA cleared after curative surgery (Fig. 6b).

Pair-wise comparison of three different RFS prediction models using different combinations of clinical features, tumor volume, imaging subtypes, ctDNA detection at pre-surgery, and ctDNA clearance status after SOC therapy is shown in Fig. 6c. We observed that the performance of the Cox model using combinations of ctDNA status along with clinical features, tumor volume and imaging subtypes achieved the optimal prediction with C-index = 0.82.

**Radiogenomics analysis identifies molecular pathways correlated to imaging subtypes**

Leveraging the available transcriptomic data from RNA sequencing, we explore molecular features associated with the radiomics subtypes using meta-GSEA for the combined datasets of all 3 cohorts in the study. The top 10 Hallmark pathways differentially enriched between high- and low-risk habitat subtypes are provided in Fig. 6d.

The top-ranked pathways downregulated in high-risk but not low-risk groups include interferon α and interferon γ responses, with angiogenesis and epithelial-mesenchymal transition consistently upregulated in high-risk tumors of the studied cohorts (Supplementary Fig. 9a, b).

## Discussion

In this multi-cohort study, we developed a habitat imaging framework to identify phenotypically distinct intratumoral subregions (habitats) through integrated analysis of CT and ¹⁸F-FDG PET. The imaging signatures extracted from these intratumoral subregions showed robust performance in stratifying lung cancer patients into 3 clinically meaningful subtypes with distinct prognoses. These imaging subtypes offer independent prognostic information beyond established clinicopathological risk factors and blood ctDNA. Furthermore, we pinpointed molecular features underlying these imaging subtypes via radiogenomics analyzes. Altogether, we have clinically and biologically validated the habitat imaging subtypes in multiple independent cohorts, including prospective trial and retrospective cohorts. However, our study is a proof-of-concept study, which lays the groundwork for future investigation to establish the robustness and clinical applicability of these identified subtypes.

While clinical, pathologic, and lifestyle factors can inform the risk of recurrence, they fail to account for patient-level dynamic evolution and inter-individual variations. We have demonstrated that the habitat radiomics model can potentially serve as an upfront earlier indicator of recurrence than standard clinical examination or follow-up scans. So, patients at an elevated risk of recurrence may be identified for potential treatment intensification, closer post-treatment monitoring, and potential enrollment in experimental protocols for more aggressive systemic management.

This is a multi-center PET/CT habitat radiomic study, emphasizing intratumor heterogeneity by dividing the gross tumor into subregions and analyzing their spatial interactions, which we performed in NSCLC patients with early-stage disease. Prior studies captured image features from the entire tumor volume but failed to account for intratumoral heterogeneity[10,22–24]. Our study has several strengths. One key aspect of our study is the integrated analyzes of CT and ¹⁸F-FDG PET built on our previous pipeline[19], whereas most prior studies focused on a single modality. In contrast to conventional radiomic studies that extract hundreds of radiomic features with feature selection to build a

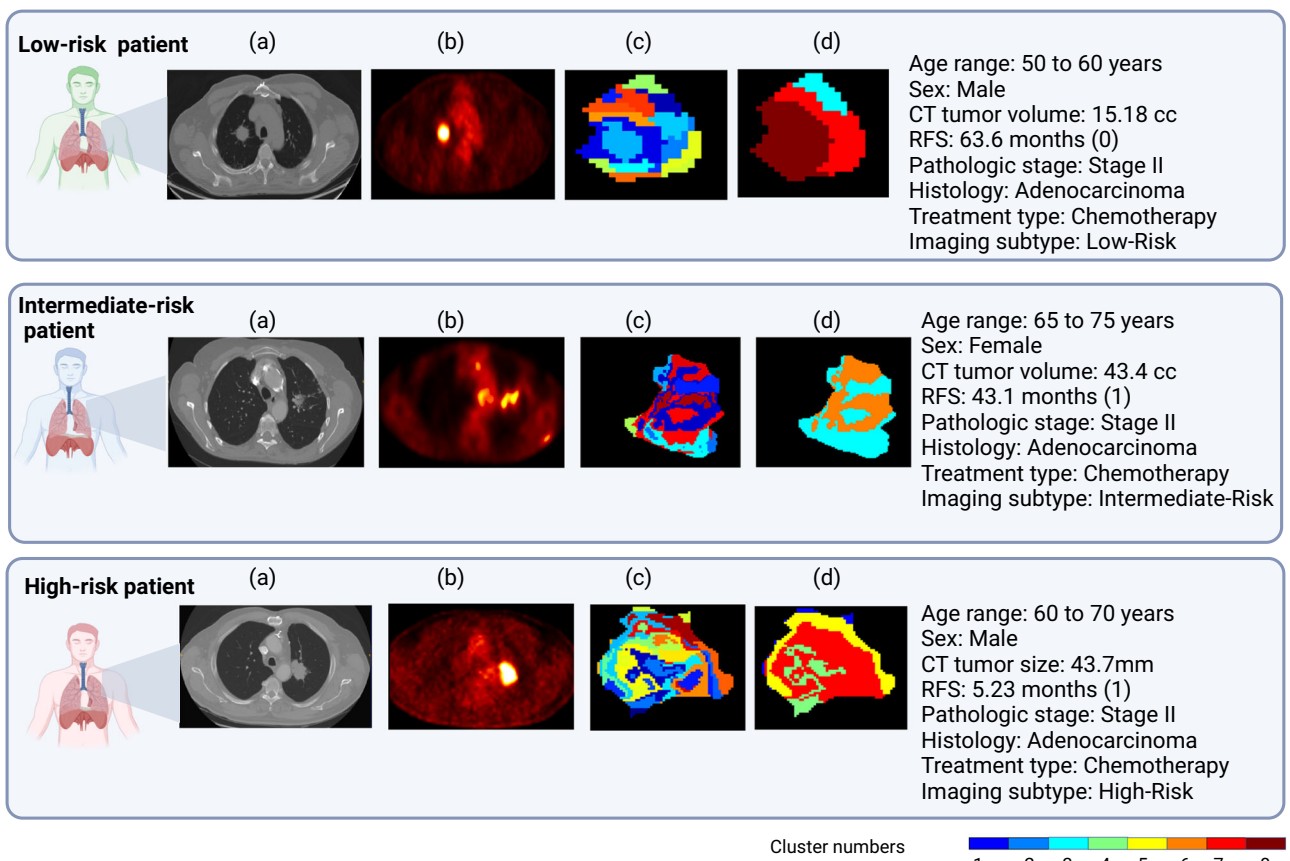

**Fig. 5 | Workflow of habitat imaging–based intratumor partitioning on chest CT and PET images.** Example patients with similar baseline clinical characteristics in the low-risk, intermediate-risk, and high-risk subtype groups. **a** CT and **b** PET scans. **c**, **d** are the fused 3D tumor volume oversegmented to superpixels and integrated by clustering using the Louvain algorithm to identify the habitat regions. Created with BioRender.com.

parsimonious model, we applied unsupervised clustering to define imaging subtypes based on the features extracted from tumor subregions. Technically speaking, the clustering approach capturing inner data structure is less prone to overfitting and more robust to small-scale and heterogenous cancer populations, similar to previous studies[10,25,26]. CT and PET scans were collected from different centers, with different scanners, imaging protocols, and reconstruction platforms. Image standardization and radiomic feature harmonization were performed to ensure more generalizable modeling. The consistent clinical performance in the external validation cohort from multi-center trial corroborated the robustness of proposed habitat imaging pipeline.

The availability of in-house gene expression data, blood-derived ctDNA, and high-quality metadata enabled us to identify the biology behind the high-risk imaging subtype, reinforcing why this subtype was associated with a higher rate of recurrence. Intriguingly, a synergistic effect was observed between ctDNA and habitat imaging subtypes, supporting the orthogonal information offered by imaging similar to our previous study[27]. If prospectively validated, these predictors can be combined to stratify NSCLC patients non-invasively, potentially identifying high-risk patients who might benefit from more intensive therapy and monitoring, while allowing for de-escalation of treatment in low-risk patients.

EMT, one of top dysregulated pathways in the high-risk subtype, is known to be correlated with lung cancer progression, metastasis, drug resistance and immune evasion[28]. Activation of EMT involves a set of molecular processes related to tumor heterogeneity. These processes include remodeling of extracellular matrix, lost adhesion of tumor cells to extracellular matrix and to each other, their mobility, and

differentiation and activation of the stemness phenotype[29]. Significant negative correlation between EMT and IFN-γ signaling revealed in the high risk tumor group have been demonstrated in lung cancer experimentally and were linked to impaired immunosurveillance in lung cancer cells[30]. It was shown that a combined treatment with IFN-γ and an SHP2 inhibitor induced enhanced anticancer activities in cell line-derived xenograft models. According to the present study, lung tumors classified as high risk by the proposed habitat imaging will have most active EMT phenotype and significant downregulation of IFN-γ signaling that make them most suitable for a combined treatment with IFN-γ and an SHP2 inhibitor.

Our study has several limitations. First, though habitat imaging has demonstrated clinical and biological value in retrospective sets, the findings need to be prospectively validated in large datasets[31] before they can transition to inform risk stratification of early-stage NSCLC. More importantly, future mechanistic studies are needed to validate these radiogenomics correlates. In addition, the habitat imaging requires both PET and CT scans that can limit its clinical application especially in screening setting. Also, we only focused on the primary tumor, and future radiomics study can also consider involved lymph nodes with demonstrated clinical values[7,32]. Second, we performed the habitat analysis by mixing different histologic subtypes (adenocarcinoma and squamous cell carcinoma) to increase the statistical power. Future studies are needed to optimize the analysis in each subtype. Third, though our habitat analysis is biology-relevant, it relies on hand-crafted pipelines, especially for the features to quantify intratumor heterogeneity. Deep-learning approaches to automate the tumor profiling will be implemented in large-size cohorts in the future to integrate multimodal data[9,33–37].

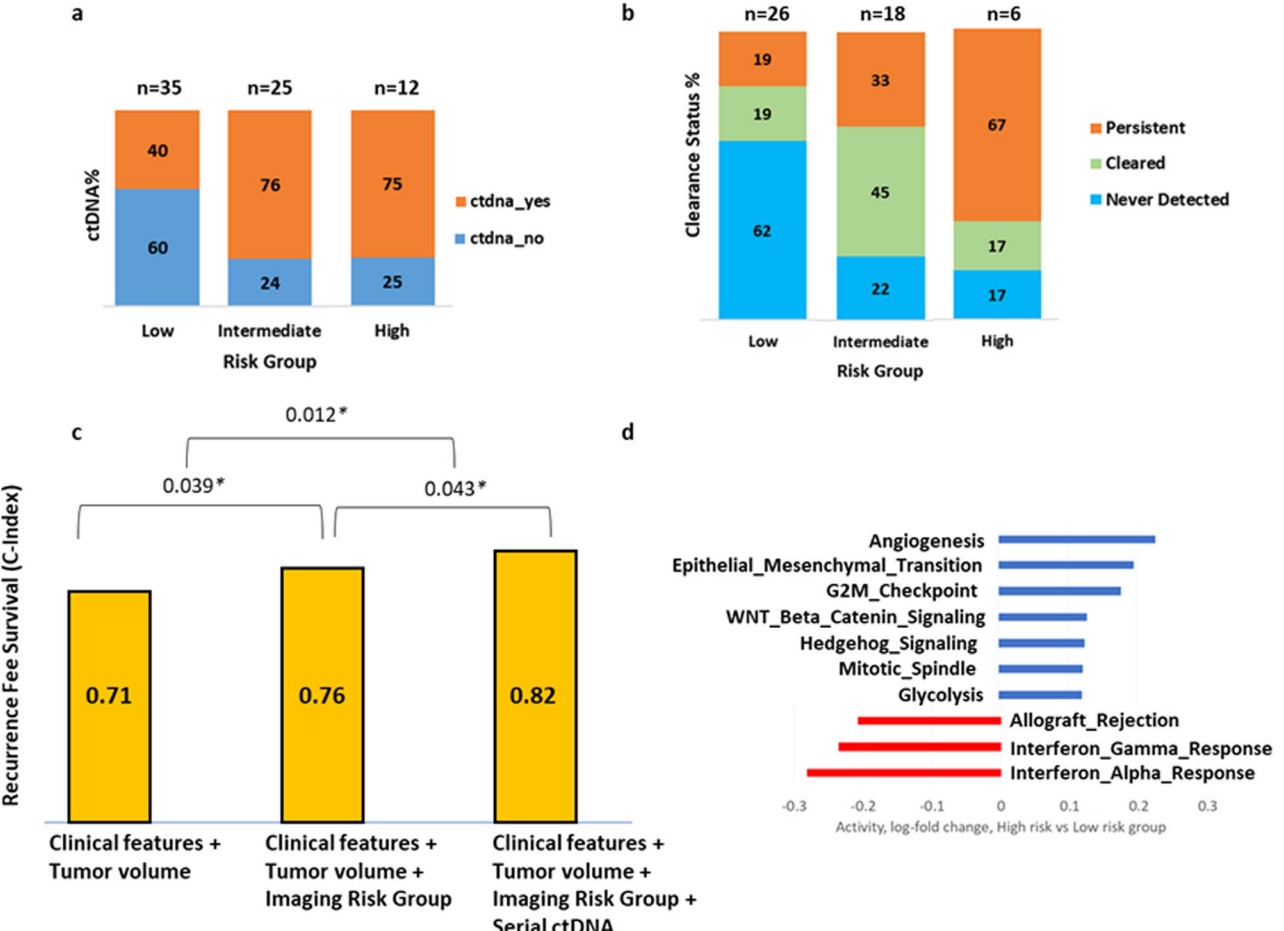

**Fig. 6 | Circulating tumor DNA (ctDNA) and gene set enrichment analysis. a** Bar plot showing the percentage of patients with ctDNA detected before surgery, stratified across different risk groups. **b** Bar plot showing the percentage of patients with ctDNA clearance status (Persistent, Cleared, Never Detected) after SOC therapy, stratified across different risk groups. **c** C-index comparison of three different RFS prediction models using combinations of clinical features, tumor volume, imaging subtypes, and serial ctDNA (ctDNA detection at pre-surgery and ctDNA clearance status after SOC therapy). **d** Top 10 Hallmark Pathways differentially active in high- versus low-risk group according to the meta-analysis of the studied cohorts. Source data are provided as a Source Data file.

In conclusion, we have developed and validated a preoperative CT and $^{18}$F-FDG PET–based signature that serves as a prognostic biomarker for patients with NSCLC. If prospectively validated, the proposed imaging signature could be used to refine individualized therapeutic selection in lung cancer.

## Methods
### Patient cohorts
This study was approved by the Institutional Review Board and compliant with the Health Insurance Portability and Accountability Act. Patient consent is waived for this retrospective study. The study cohorts are shown in Fig. 1. The first cohort was selected from the PROSPECT cohort (Profiling of Resistance Patterns and Oncogenic Pathways in Evaluation of Cancers of the Thorax) from The University of Texas MD Anderson Cancer Center[38–40]. From this group, we included 75 patients with available pre-surgery CT and PET scans. The second cohort was selected from TCIA (The Cancer Imaging Archive)[41,42]. Out of 211 patients with histologically or cytologically confirmed stage I to IIIB NSCLC with surgical treatment at Stanford Medical Center, we included 124 patients with pre-surgery CT and PET scans. The third cohort was the ICON (ImmunogenomiC prOfiling of Non-small cell lung cancer Project)[43]. From this group of prospectively enrolled patients with clinical stage IA–IIIA NSCLC prior to resection, we included 133 patients with available pre-surgery CT and PET scans. The fourth cohort was from the multi-center ACRIN 6668/RTOG 0235

trial[44,45]. After image and clinical data collection, 62 patients were included in our study for quantification using habitat imaging analysis. PROSPECT and TCIA cohorts were used for model discovery, with ICON and ACRIN sets for internal and external validation. The demographic and clinical characteristics of the study population are presented in Table 1.

### Study design
The three main stages of the habitat imaging–based radiomics analysis are shown in Fig. 2's overall study design: data acquisition and pre-processing, habitat imaging analysis, and clinical and biological value of imaging subtypes. In brief, we collected and quality-controlled the multicenter CT and PET scans at baseline. Habitat imaging was used to define the tumor subregions and subsequently to profile the architecture and interaction of these subregions. Then, using an unsupervised clustering approach, patients were stratified into habitat imaging subtypes. Finally, we assessed the clinical value of the imaging subtype in relation to conventional CT/PET markers, clinicopathologic factors, and blood ctDNA for predicting RFS and OS. Furthermore, we investigated the molecular pathway level correlates of imaging subtype through radiogenomics[46].

### Image acquisition
Firstly, PROSPECT images were obtained using a dedicated CT/PET system (Discovery ST, STe, or RX; GE Medical Systems and GE

Healthcare) with slice thickness 2.5 to 10 mm (median: 5 mm). After fasting for a minimum of 6 h and undergoing confirmation of a blood glucose level less than 200 mg/dL, patients received an intravenous infusion of 259–740 MBq 18F-FDG. A CT scan was performed for anatomic correlation and attenuation correction; PET images were subsequently obtained 60–90 min after 18F-FDG infusion[47].

Secondly, images, gene expression, and clinical information were publicly available at TCIA. TCIA dataset has CT images in DICOM format retrospectively collected using different scanners, scanning protocols and parameters: slice thickness of 0.625-3 mm (median:1.5 mm). Fasting Fluorodeoxyglucose $^{18}$F-FDG PET/CT data were collected using GE Discovery PET/CT scanner. FDG Dose and uptake time were 138.90–572.25 MBq (mean 309.26 MBq) and 23.08–128.90 min (mean 66.58 minutes), respectively.

Thirdly, ICON CT scans were acquired in a multidetector scanner following IV contrast administration unless contraindicated. Multiplanar CT image series were reconstructed with 2.5 mm slice thickness using standard and high spatial reconstruction algorithms. FDG-PET/CT imaging was performed using Discovery STE PET/ CT scanner (GE Healthcare, Waukesha, WI, USA). All patients fasted for 6 h before the FDG injection and had confirmed normal fasting blood glucose level of less than 200 mg/dL. PET was performed in three-dimensional mode at 3–5 min per bed station depending on patient BMI. The acquired PET data were corrected for scatter coincidences, random coincidences, deadtime, and attenuation and reconstructed using OSEM on standard vendor-provided workstations. Non-contrast-enhanced CT images from the base of the skull to the mid-thigh were acquired in helical mode (speed, 13.5 mm per rotation) during shallow breathing at a 3.75 mm slice thickness, a tube voltage of 120 kVp, and 0.5 s rotation with tube current modulation[48].

### Harmonization of multi-site CT and PET scans
Given the variations of imaging scanners and protocols, we crafted an image pre-processing pipeline (Fig. 2a) to harmonize these scans. We normalized the CT scans with lung windowing to highlight anatomical structures. We also normalized PET scans by body weight (SUVbw) into standardized uptake value (SUV) map to quantify FDG uptake and remove variability. We then spatially aligned the PET images and the CT using the affine registration in Elastix software (version 5.0.1). Moreover, the quality of the registration was visually inspected for alignment across the tumor region, ipsilateral lung, and other tissues. Manual alignment using 3D Slicer (version 4.11) was performed for cases that needed the registration outcome fine-tuned. We then computed the local entropy of the normalized CT and PET scans and ensure that the multimodal images are normalized to identical pixel resolution of 1 mm². The identification and segmentation of target tumor regions were meticulously conducted by our experienced clinical radiologists, drawing upon their expertize and the available clinical datasets. The precise annotation of tumor regions was executed using MIM software, employing a manual segmentation process. In order to guarantee accuracy and consistency, the segmented results underwent a comprehensive cross-checking process conducted by three independent and expert radiologists. This rigorous approach was taken to ensure the reliability and precision of our tumor region annotations.

### Habitat imaging framework
To distinguish spatially distinct subregions within the tumor region, we proposed a habitat imaging framework (Fig. 2b) by extending our previous computational habitat framework[19,49–51]. For further analysis, the spatially aligned tumor regions created from CT and PET data were used. In addition, the complexity of the local texture was calculated using local entropy maps of the tumor. Entropy is a statistical measure of randomness or uncertainty that can be used to characterize the texture of an image. A greater entropy value denotes deeper features,

and an entropy map is a representation of the entropy values shown as a grayscale image. The CT, CT entropy, PET, and PET entropy images were fused to integrate the anatomic, metabolic, intensity, and texture information to define tumor heterogeneity. By image fusion, we meant adding the corresponding pixel values of the different image modalities (CT, CT entropy, PET, and PET entropy) to generate texture rich composite images used for habitat detection. Two steps of clustering were used to split the tumor region. First, the feature maps were over-segmented into superpixels at the patient level using the k-means clustering algorithm with Euclidean distance as the similarity metric. Based on gray-level frequency distribution from the histogram analysis, first-order features ($n = 40$) were extracted for individual superpixel, including skewness, kurtosis, mean, median, first quartile, second quartile, interquartile range, standard deviation, variance, and energy, calculated separately on 4 different maps. Second, using the Louvain algorithm in Seurat (version 4.0), we clustered the 3 cohorts separately. After confirming the consistency of clustering results across different cohorts, we aggregated superpixels from the entire population level to identify the unified habitat regions.

### Quantification of intratumor heterogeneity
To quantify the intra-tumor heterogeneity on the habitat maps, we used the multiregional spatial interaction (MSI) matrix[49,50], which summarizes the spatial co-occurrence statistics among different habitat regions (Fig. 2b). In detail, for every tumor voxel, the co-occurring pairs of its neighbors were scanned and added to the corresponding cells in the MSI matrix. By co-occurring pairs, we meant that we checked each pixel and find its immediate nearest neighbors in the 3D. Pixels are connected if their edges or corners touch. After this process was iterated through all tumor voxels, the spatial distribution and interaction of the intratumoral habitats were abstracted in the MSI matrix. In order to quantify intratumoral spatial heterogeneity, quantitative features were then extracted from the MSI matrix. These features included 1) the first-order statistical features for the absolute volume of each subregion, the proportion of each subregion, and borders of 2 interacting subregions; and 2) the second-order statistical features of contrast, homogeneity, correlation, and energy computed on the normalized MSI matrix. After constructing the MSI matrix based on the pixel connectivity, we consider the MSI matrix as input to the graycoprops function in MATLAB to calculate 4 second-order statistical properties of the MSI matrix including contrast, homogeneity, correlation, and energy.

### Imaging subtype identification and validation
Given the MSI features, we applied a consensus clustering algorithm to identify the optimal number of imaging subtypes in the discovery and validation cohorts separately. Compared to the one-time clustering of conventional algorithms, consensus clustering repeats clustering on subsampled patients and features and is more robust for detecting intrinsic clusters. We selected the partition-around-medoids clustering algorithm with the Spearman distance metric. The cluster number varied from 2 to 5, and an optimal cluster number that produced optimal consensus was chosen. The consensus matrix from k = 2 to k = 5 was tested using the cumulative distribution function (CDF) curve. A perfect consensus matrix would be filled with 0 and 1 only, thus showing an ideal step function of the CDF curve.

### Clinical evaluation of imaging subtypes for prognostic significance
We evaluated the prognostic capacity of the identified imaging subtypes (Fig. 2c). In univariate analysis, we compared imaging subtype with established clinicopathological risk factors (including age, sex, clinical T category, smoking status, histology, and neoadjuvant chemotherapy) and conventional CT/PET metrics (tumor volume delineated on the CT scan, metabolic tumor volume [MTV], total lesion

glycolysis [TLG], and SUVmax). Then, we tested the added clinical value of the imaging subtypes over that of these known risk factors using multivariate analysis. Tumor volume stands out as a pivotal predictor in lung cancer prognosis, exerting a substantial influence on disease recurrence and overall survival[52–54]. Acknowledging its critical role, we systematically address the potential impact of tumor volume on the significance of our proposed habitat imaging subtypes[53].

## Integrated analysis of ctDNA and imaging subtypes

The presence of ctDNA at pre-surgery and after definitive standard-of-care (SOC) therapy indicates residual micro-metastatic disease, which is associated with poor RFS, whereas its absence predicts a low risk of recurrence for NSCLC after surgery[55–57]. After definitive standard-of-care (SOC) therapy is defined as surgery and neoadjuvant or adjuvant chemotherapy, and/or post-operative radiotherapy (PORT) if administered. Based on ctDNA clearance status, three categories were defined: without detectable circulating tumor (ctDNA) throughout the study (Never Detected group), patients who cleared ctDNA during treatment (Cleared group), and patients with persistent ctDNA detectable (Persistent group). Given ctDNA presence/clearance status, we evaluated the prognostic significance of ctDNA combined with our proposed imaging subtypes in the validation cohort and explored whether they can be complementary to each other.

## Identifying molecular pathways correlated with imaging subtypes

We performed molecular pathway analyzes using paired transcriptomic data from three studied cohorts of patients to explore biological mechanisms underlying the habitat imaging subtyping. Gene set enrichment analysis was applied for each individual cohort and meta–gene set enrichment analysis for the combined cohorts using R package QuSAGE. We focused on the hallmark gene set (version 2022.1) from the Molecular Signatures Database (MSigDB). A false-discovery-rate of <0.2 was used to select significantly activated and suppressed pathways.

## Statistical analysis

Kaplan-Meier analysis and the log-rank test were used to evaluate patient stratification into different risk groups in terms of the endpoints (RFS and OS). Univariate and multivariate analysis using the Cox proportional hazards model was performed to correlate different risk predictors and endpoints. The hazard ratio was used to measure the degree of survival differences. All statistical tests were 2-sided, with $P < 0.05$ indicative of a statistically significant difference. All statistical analyzes were performed in R (version 4.1.2).

## Reporting summary

Further information on research design is available in the Nature Portfolio Reporting Summary linked to this article.

## Data availability

The source data for Figs. 3, 4, 6 and Tables 1, 3, 4 are provided with this paper. The supplementary data for Supplementary Figs. 1-9 and Supplementary Tables 1-10 are also provided. The FDG-PET/CT and clinical data of TCIA cohort are publicly available on The Cancer Imaging Archive https://wiki.cancerimagingarchive.net/display/Public/NSCLC +Radiogenomics#28672347d6e83195f69f438ca0d1a3d20fbc450d. The raw FDG-PET/CT and clinical data of ACRIN 6668/RTOG 0235 cohort are publicly available on The Cancer Imaging Archive https://wiki.cancerimagingarchive.net/pages/viewpage.action?pageId= 39879162#398791626e061ab3228446d59c8ce2ac2d1aa117. The raw FDG-PET/CT of ICON and PROSPECT are not publicly shared to protect patient privacy, but are available for research use from the corresponding author. MTA is required to be approved by MD Anderson committees by providing the research plan and is restricted to non-

commercial academic research purposes. Request can be submitted to J.W. and will receive an internal review response within 30 days. In addition, anonymized data and the input for the predictive models are available at Zenodo https://doi.org/10.5281/zenodo.10611536[58]. Deidentified ctDNA data for patients in the internal validation cohort are available in Source data for Fig. 6. The genomics data of PROSPECT cohort are available at GEO repository GSE42127. The genomics data of TCIA cohort are available at GEO repository GSE103584. The genomics data of ICON cohort is hosted by The European Bioinformatics Institute (EBI) and the Centre for Genomic Regulation (CRG) under the accession code: EGAD50000000361. Source data are provided with this paper.

## Code availability

The code in this study has been deposited in the repository available at https://doi.org/10.5281/zenodo.10611536[58]. The Habitat Imaging code is available at https://github.com/WuLabMDA/Habitat-Analysis.

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

## Acknowledgements

This work was supported by the generous philanthropic contributions to The University of Texas MD Anderson Lung Moon Shot Program and the MD Anderson Cancer Center Support Grant P30CA016672. This work was supported by the Tumor Measurement Initiative through the MD Anderson Strategic Initiative Development Program (STRIDE). This research was partially supported by the National Institutes of Health (NIH) grants R00CA218667 (J.W.), R01CA262425 (T.C. and J.W.), R01CA276178 (N.I.V. and J.W.). This work was supported by generous philanthropic contributions from Mrs. Andrea Mugnaini and Dr. Edward L.C. Smith. This work was supported by Rexanna's Foundation for Fighting Lung Cancer. J.B is a CPRIT CURE trainee (RP210028). The manuscript was edited by Sarah Bronson, ELS, of the Research Medical Library at MD Anderson.

## Author contributions

S.J.S., M.A., J.W. conceived and designed the study. S.J.S., M.B.S., M.S., J.D.B., M.Q., J.M.G., L.H., M.C.B.G. collected and annotated data. S.J.S., M.A. and J.W. developed habitat imaging methodology. S.J.S., T.V.K. and J.W. performed radiogenomics analysis. S.J.S., M.A., T.V.K. and J.W. prepared the first draft of the manuscript. All authors reviewed, revised, discussed results and contributed to the finalization of the manuscript. All authors read and approved the final version of the manuscript, and were responsible for the final decision to submit the manuscript for publication.

## Competing interests

X.L. reports Consulting/advisory fees from Eli Lilly, EMD Serono (Merck KGaA), AstraZeneca, Spectrum Pharmaceutics, Novartis, Regeneron, Boehringer Ingelheim, Hengrui Therapeutics, Bayer, Teligene, Taiho, Daiichi Sankyo, Janssen, Blueprint Medicines, Sensei Biotherapeutics, SystImmune, ArriVent, Abion, and Abbvie Research Funding to Institution from Eli Lilly, EMD Serono, ArriVent, Dizal, Teligene, Regeneron, Janssen, ThermoFisher, Takeda, and Boehringer Ingelheim. Travel Support from EMD Serono, Janssen, and Spectrum Pharmaceutics. Stock options from BlossomHill. T.C. reports speaker fees and honoraria from The Society for Immunotherapy of Cancer, Bristol Myers Squibb, Roche, Medscape, and PeerView; having an advisory role or receiving consulting fees from AstraZeneca, Bristol Myers Squibb, EMD Serono, Merck & Co, Genentech, and Arrowhead Pharmaceuticals; and institutional research funding from AstraZeneca, Bristol Myers Squibb, and EMD Serono. N.I.V. receives consulting fees from Sanofi, Regeneron, Oncocyte, and Eli Lilly; and research funding from Mirati outside the submitted work. J.Y.C. reports research funding from BMS-MDACC, Siemens Healthcare, and consultation fees from Legion Healthcare Partners. L.Y. has grant support from Lantheus Inc. M.C.B.G. has received research funding from Siemens Healthcare. I.W. has received honoraria from Genentech/Roche, Astra Zeneca, Merck, Guardant Health, Flame, Novartis, Sanofi, Daiichi Sankyo, Dava Oncology, Amgen, GlaxoSmithKline, HTG Molecular, Jansen, Merus, Imagene, G1 Therapeutics, Abbvie, Catalyst Therapeutics, Genzyme, Regeneron, Oncocyte, Medscape, Platform Health, Pfizer, Physicians' Education Resource, HPM Education, and Aptitude Health; Additionally, I.W. has received research support from Genentech, Merck, Bristol-Myers Squibb, Medimmune, Adaptive, Adaptimmune, EMD Serono, Pfizer, Takeda, Amgen, Karus, Johnson & Johnson, Bayer, Iovance, 4D, Novartis, and Akoya. D.L.G. has served on scientific advisory committees for Menarini Ricerche, 4D Pharma, Onconova, and Eli Lilly and has received research support from Takeda, Astellas, NGM Biopharmaceuticals, Boehringer Ingelheim and AstraZeneca. J.V.H. reports being on scientific advisory boards for AstraZeneca, Boehringer Ingelheim, Genentech, GlaxoSmithKline, Eli Lilly, Novartis, Spectrum, EMD Serono, Sanofi, Takeda, Mirati Therapeutics, BMS, and Janssen Global Services; receiving research support from AstraZeneca, Takeda, Boehringer Ingelheim, and Spectrum; and receiving licensing fees from Spectrum. C.C.W reports research support from the Medical Imaging and Data Resource Center from NIBIB/University of Chicago and royalties from Elsevier. J.Z. reports serving on the consulting/advisory board of Bristol-Myers Squibb, AstraZeneca, Novartis, Johnson & Johnson, GenePlus, Innovent, Varian, and Catalyst; receiving research grants to institutions from Merck, Novartis, and Johnson & Johnson. J.W. reports research funding from Siemens Healthcare. The remaining authors declare no competing interests.

## Additional information

¹Department of Imaging Physics, The University of Texas MD Anderson Cancer Center, Houston, TX, USA. ²Department of Genomic Medicine, The University of Texas MD Anderson Cancer Center, Houston, TX, USA. ³Department of Biomedical Engineering, Duke University, Durham, NC, USA. ⁴Michigan Center for Translational Pathology, University of Michigan, Ann Arbor, MI, USA. ⁵Natural Sciences, Rice University, Houston, TX, USA. ⁶Department of Thoracic and Cardiovascular Surgery, The University of Texas MD Anderson Cancer Center, Houston, TX, USA. ⁷Department of Thoracic/Head and Neck Medical Oncology, The University of Texas MD Anderson Cancer Center, Houston, TX, USA. ⁸Department of Pulmonary Medicine, The University of Texas MD Anderson Cancer Center, Houston, TX, USA. ⁹Department of Nuclear Medicine, The University of Texas MD Anderson Cancer Center, Houston, TX, USA. ¹⁰Department of

Thoracic Imaging, The University of Texas MD Anderson Cancer Center, Houston, TX, USA. [11]Department of Radiation Oncology, The University of Texas MD Anderson Cancer Center, Houston, TX, USA. [12]Institute of Data Science in Oncology, The University of Texas MD Anderson Cancer Center, Houston, TX, USA. [13]Department of Radiation Physics, The University of Texas MD Anderson Cancer Center, Houston, TX, USA. [14]Department of Translational Molecular Pathology, The University of Texas MD Anderson Cancer Center, Houston, TX, USA. [15]Department of Biostatistics, The University of Texas MD Anderson Cancer Center, Houston, TX, USA. [16]Lung Cancer Genomics Program, The University of Texas MD Anderson Cancer Center, Houston, TX, USA. [17]Lung Cancer Interception Program, The University of Texas MD Anderson Cancer Center, Houston, TX, USA. [18]These authors contributed equally: Sheeba J. Sujit, Muhammad Aminu. [19]These authors jointly supervised this work: Jianjun Zhang, Tina Cascone, Jia Wu. ✉e-mail: jwu11@mdanderson.org

