## [Peer Review File · Nature Communications]

Unleashing PET/CT Habitat Imaging Potential: Elevating Recurrence Prediction in NSCLC with ctDNA and Radiogenomics InsightsREVIEWER COMMENTS

Reviewer #1 (Remarks to the Author): expert in machine learning and imaging

This is a well designed study with suggestive if not definitive results. The habitat imaging framework to measure intra-tumoral heterogeneity and define imaging subtypes is a significant contribution to the field. The approach to the discovery of regionalized radiomic signatures is well founded.

The results are highly suggestive but as the authors argue, additional evidence is needed to confirm their key biological findings.

A few minor issues were identified which may have an impact on reproducibility:

The authors did not indicate how target tumor regions were identified and segmented.

A publicly available TCIA data set was used but not clearly identified by referencing its DOI. It is possible this dataset contained RNAseq data but it is unclear if these data were used.

It appears ctDNA was only available from proprietary data sets, but it is unclear which data set(s) contained these data. Are these data from the same subjects as the imaging data?

How large are the ctDNA and RNAseq data sets?

I strongly urge publication after these minor issues are addressed.

Reviewer #2 (Remarks to the Author): expertise in non-small cell lung cancer radiomics

The manuscript describes a FDG PET/CT-based habitat imaging analysis that classifies tumors into subtypes prognostic for disease recurrence and overall survival in resected non-small cell lung cancer. The subtypes exhibit synergistic effects with ctDNA and are associated to specific molecular pathways. While spatial heterogeneity radiomics is a promising technique, the robustly validated independent prediction power of the proposed tumor subtyping could be questioned based on current analyses:

1. As in most radiomics work, e.g. the seminal Aerts et al. study, the major concern is the independence of the radiomic signature (in this case the habitat-based tumor subtyping) from the simple parameter of tumor volume (Vallières et al. *Journal of Nuclear Medicine* 2018; 59:640, Welch et al. *Radiotherapy and Oncology* 2019). Tumor volume is one of the strongest known predictors of disease recurrence and overall survival in lung cancer and it could thus bias the proposed habitat imaging subtypes significance. Extensive safeguards should be applied to rule this out.

1.1 A multivariable analysis including tumor volume was presented (Table 3), however, only in the integrated cohort (discovery and validation sets combined). Moreover, 12 variables were included which is potentially problematic for generalizability given the number of events. The independent impact of habitat imaging subtypes should be analyzed more in detail as it is the main message of this work. By separate multivariable analyses in the discovery and validation cohorts. Univariable significance and/or exclusion of intercorrelated variables may guide variable selection.

1.2 In Supplementary Figures 5-6 Kaplan-Meier curves depict the impact of the habitat imaging subtypes in groups with similar clinicopathological parameters. It is advised to similarly analyze groups formed by tumor volume. Is the habitat imaging subtyping as useful in small as in large tumors?

1.3 The correlation of simple tumor-related parameters as tumor volume, MTV, TLG, etc. on the one hand with the habitat imaging subtypes on the other hand should be investigated.

1.4 The correlations of tumor volume with the ctDNA data and with the molecular profile findings should be checked to eliminate the risk of these being just a surrogate of tumor volume.

2. Some additional known risk factors of disease recurrence and/or overall survival should be included in the univariable, multivariable and correlation analyses. For instance primary tumor

location categories (e.g. Tonneau et al. Radiotherapy and Oncology 2023) and lung side/lobe location could potentially be correlated to the habitat imaging subtypes. This potential bias should be excluded.

It is not clear how many patients received postoperative radiotherapy. Radiotherapy-related parameters are important risk factors of the studied endpoints: prescription dose, but also tumor location (associated to heart dose exposure) were linked to overall survival after radiotherapy. Radiotherapy-related parameters should thus at least be reported and their correlations with outcome and habitat imaging subtypes investigated.

3. A comparison of the habitat-based tumor subtype classification with a classical radiomics approach to tumor classification (directly calculating features from the whole tumor volume) could enhance the importance of proposed technique. Does radiomics using habitat imaging perform significantly better?

4. The validation dataset was collected from the same institution as a large part of the discovery dataset. No true external validation of the technique was thus performed. Moreover, no firm prediction model (with Cox model coefficients, MSI feature cut-offs to be used, etc.) has been reported nor validated.

Specific comments:

Line 148: What was the treatment period in the different cohorts?

Line 212: What was the final pixel resolution? The same resolution for every patient?

Line 221: 'images were fused', what does this exactly mean?

Line 237: 'co-occurring pairs of its neighbors', please elaborate. Neighbors in 2D or 3D?

Line 243: How were 'second-order statistical features' derived from the MSI matrix?

Line 260: 'tumor volume', meaning primary gross tumor volume as delineated on CT, in cc?

Line 298: What was the median follow-up time and the median OS in the different cohorts?

Line 305: 'CT is high', what does 'high' mean here?

Line 381: 'can be accurately identified', this statement is currently not supported by data. How is 'accurately' defined, and which accuracy metric was calculated?

Ref. 27 is not referenced.

Ref. 42 is incomplete.

Table 2: How are the MSI matrix-based features that are referred to as 'interactions between subregions and border' defined? Is the border one of the 8 subregions listed in the MSI matrix?

Table 3: Please add the units of the variables in order to interpret HR, e.g. tumor volume in cc?

Figure 5: Tumor volume should be reported instead of tumor diameter size.

Figure 6c: Are the clinicopathological features included in these prediction models? Is the difference between the c-indexes significant? It would additionally be helpful to report the improvement in c-index when adding the habitat imaging risk groups to a baseline clinicopathological feature model (including tumor volume).

Suppl. Fig.1: Do the cohorts come independently to the conclusion of splitting into 8 clusters or is there cross-talk between discovery and validation cohorts in the methodology on this point? Line 231 'we further confirmed the consistency of these subregions across the 3 cohorts', how was this achieved?

Altogether, was CT or PET data most informative in the developed habitat image subtype signatures?

Suppl. Fig. 4c: No significance result is mentioned for MSI 19 in the discovery cohort.

Suppl. Fig. 7b: What could be the reason for the different behavior observed in the validation cohort for these 3 molecular pathway features?

Reviewer #3 (Remarks to the Author): clinical expertise in non-small cell lung cancer

Thanks for including me to review this manuscript "Habitat Imaging Subtypes Defined on CT and 18F-FDG PET Complement Circulating Tumor DNA in Predicting Disease Recurrence and Correlate with Distinct Molecular Pathways in Resected Non-Small Cell Lung Cancer". This study focused on intratumor heterogeneity by dividing the gross tumor into subregions and analyzing spatial interactions, which were performed in NSCLC patients with early-stage disease. Moreover, the availability of in-house gene expression data, blood-derived ctDNA, and high-quality metadata enabled us to identify the biology behind the high-risk imaging subtype, reinforcing why this subtype was associated with a higher rate of recurrence. Overall, this is a nice study that aimed to

build a habitat imaging framework in predicting recurrence-free survival (RFS) following curative surgery. However, there are still some minor concerns about this frame work. First, in clinical practice when patients received CT scan and showed ground glass opacity less than 8mm, this subgroup of patients may not receive PET-CT later. They would receive biopsy, surgery or just follow up. So this frame work which was based on both CT and PET might somehow limit the amplification? Furthermore, is there any genomic data that might be associated with predict outcomes such EGFR mutation/ALK fusion?

RESPONSE TO REVIEWERS' COMMENTS

Reviewer #1: This is a well-designed study with suggestive if not definitive results. The habitat imaging framework to measure intra-tumoral heterogeneity and define imaging subtypes is a significant contribution to the field. The approach to the discovery of regionalized radiomic signatures is well founded.

Reply: We thank the reviewer for the overall favorable comments and remarks on our work. We appreciate the time and effort you dedicated to evaluating our manuscript. We have addressed your suggestions as follows.

1. The results are highly suggestive but as the authors argue, additional evidence is needed to confirm their key biological findings.

Reply: We sincerely appreciate your insightful comment emphasizing the need for additional evidence to support the key biological findings outlined in our study. One notable strength of our research is the execution of radiogenomics analysis in three distinct, robust and diverse cohorts, each comprising paired imaging and RNA sequencing data (**Fig 2c**). Importantly, the consistent identification of biological findings across these independent sets bolsters the reliability of the radiogenomics findings.

Furthermore, we agree with you that while our findings are highly suggestive, establishing a 'mechanistic' link beyond correlative between the imaging subtypes and their biological underpinnings is imperative. It is important to note, however, that this mechanistic experiment lies beyond the scope of the current study, while we fully recognize its significance. We are committed to incorporating such experiments in our future study plans, aiming to deepen our understanding and provide a more comprehensive perspective on the identified imaging subtypes and their biological drivers. We have modified the Discussion section of the revised manuscript to incorporate this important point and future directions to address it (**page 23. Line 477-481**):

“First, though habitat imaging has demonstrated clinical and biological value in retrospective sets, the findings need to be prospectively validated in large datasets before they can transition to inform risk stratification of early-stage NSCLC. More importantly, future mechanistic studies are needed to validate these radiogenomics correlates.”

2. A few minor issues were identified which may have an impact on reproducibility: The authors did not indicate how target tumor regions were identified and segmented.

Reply: *“The identification and segmentation of target tumor regions were meticulously conducted by our experienced clinical radiologists, drawing upon their expertise and the available clinical datasets. The precise annotation of tumor regions was executed using MIM software, employing a manual segmentation process. In order to guarantee accuracy and consistency, the segmented results underwent a comprehensive cross-checking process conducted by three independent and expert radiologists. This rigorous approach was taken to ensure the reliability and precision of our tumor region annotations.”* We added these technical details to the Method section in the revised manuscript (**page 11-12, lines 216-223**).

3. A publicly available TCIA data set was used but not clearly identified by referencing its DOI. It is possible this dataset contained RNAseq data, but it is unclear if these data were used.

Reply: We thank you for pointing this out. We have added the reference for this dataset (Bakr, S. et al. Scientific Data, 2018). As the reviewer suspected, the TCIA dataset encompasses RNAseq data, with a total of 79 samples utilized in our analysis. To enhance the clarity of the datasets, we have revised **Table 1** and **Fig. 2** in the manuscript (see below). A **new sub-table** has been incorporated into **section (c) of Fig. 2** to illustrate the utilization of RNAseq data across all cohorts. This modification aims to provide a comprehensive representation of the incorporation of three paired RNAseq and CT cohorts in our radiogenomics analysis.

Table 1. Summary of demographic and clinical characteristics from the two study cohorts

Parameter	Discovery cohort (n =199)	Internal validation cohort (n = 133)	External validation cohort (n = 62)	P - value
Age				
Median (SD)	68.0 (10.6)	67.0 (9.5)	66.5 (9.25)	0.0341 [†]
Gender, n (%)				0.5753 [†]
Male	122 (61%)	63 (47%)	39 (63%)	
Female	77 (39%)	70 (53%)	23 (37%)	
P Stage (AJCC 7th ed.), n (%)				0.4751 [†]
0	4 (1.96%)	1 (0.73%)	-	
I	114 (55.88%)	50 (36.76%)	-	
II	41 (20.09%)	47 (35.33%)	2 (3%)	
III	40 (19.60%)	34 (25%)	60 (97%)	
Histology, n (%)				0.0375
Adenocarcinoma	154 (80%)	89 (67%)	-	
Squamous cell	39 (20%)	31 (23%)	-	
Other	6 (3%)	13 (10%)	-	
Smoking History, n (%)				0.1056
Smoking	157 (79%)	111 (83%)	-	
Never	42 (21%)	22 (17%)	-	
RNA Sequence, n (%)				
Available	115 (58%)	93 (70%)	-	
ctDNA, n (%)				
At baseline prior to surgery	-	72 (54%)	-	
Clearance status after surgery	-	50 (38%)	-	
Median Follow-up time (months)	50	44	20	0.07 [†]
Median OS time (months)	35	20	13	0.24 [†]
Recurrence Free Survival, n (%)				0.5658
Recurrence (1)	73 (37%)	42 (32%)	-	
No Recurrence (0)	126 (63%)	91 (68%)	-	
Overall Survival, n (%)				0.03751 [†]
Dead (1)	92 (46%)	30 (23%)	41 (72%)	
Alive (0)	107 (54%)	103 (77%)	16 (28%)	

[†] P - value was calculated using Pearson's Chi-square test comparing Discovery cohort and Integrated validation cohorts

Fig 2. Study design of habitat imaging–based radiomic analysis of NSCLC patients.

a. Overview of the data collected and pre-processing stages of the CT and ^{18}F -FDG PET images. The pre-processing involved ^{18}F -FDG PET to CT registration and fusion of segmented tumor regions from ^{18}F -FDG PET and CT images along with their local entropy maps. **b.** The habitat imaging analysis framework consisted of a 2-stage clustering process: Individual-level clustering, where tumor regions of each patient were over-segmented into superpixels; and population-level clustering, where clustering was performed on superpixels pooled from all patients to discover distinct tumor subregions. The multiregional spatial interaction (MSI) matrix summarizes the co-occurrence statistics among different tumor subregions. The 92 MSI features extracted from the MSI matrix identified patient subtypes. **c.** Radiogenomic analysis along with ctDNA metrics confirm the clinical and biological relevance of the identified imaging subtypes.

4. It appears ctDNA was only available from proprietary data sets, but it is unclear which data set(s) contained these data. Are these data from the same subjects as the imaging data?

Reply: Yes, the reviewer is correct in that the availability of ctDNA was exclusive to the ICON dataset, which served as our internal validation cohort. Notably, the ctDNA samples are derived from the same subjects as the imaging data, ensuring a direct correlation between the two modalities. Specifically, the ctDNA data includes pre-surgery information for 72 subjects and post-surgery clearance status for 50 subjects. To enhance the transparency of our methodology, we have **updated Table 1** in the manuscript. A **new table** has been incorporated into **section (c) of Fig. 2**, presenting a detailed overview of the utilization of ctDNA data available from the ICON dataset. This addition aims to provide a comprehensive representation of the integration of ctDNA data in our analysis.

5. How large are the ctDNA and RNAseq data sets?

Reply: We have **revised Fig. 2** in our manuscript to include a new table in section (c). This table has been added to explicitly depict the utilization of both ctDNA and RNAseq data across all cohorts. By incorporating this information, we aim to enhance the clarity and comprehensiveness of our findings, ensuring that the integration of ctDNA and RNAseq data is transparently represented in the visualization.

RNASeq data	ctDNA data
Discovery set (n = 115)	Internal validation set
Internal validation set (n = 93)	➤ ctDNA at baseline prior to surgery (n=72)
	➤ ctDNA clearance status after surgery (n = 50)

6. I strongly urge publication after these minor issues are addressed.

Reply: We sincerely thank you for your constructive feedback and thoughtful evaluation of our work. We genuinely appreciate your dedication to the peer-review process and your support in advancing the quality of scientific publications.

Reviewer #2: The manuscript describes a FDG PET/CT-based habitat imaging analysis that classifies tumors into subtypes prognostic for disease recurrence and overall survival in resected non-small cell lung cancer. The subtypes exhibit synergistic effects with ctDNA and are associated to specific molecular pathways. While spatial heterogeneity radiomics is a promising technique, the robustly validated independent prediction power of the proposed tumor subtyping could be questioned based on current analyses:

Reply: We appreciate your thoughtful evaluation of our manuscript and your insightful comments. We acknowledge the concern you raised regarding the robust validation of the independent prediction power of the proposed tumor subtyping in the context of current analyses. Recognizing the significance of ensuring the reliability and validity of our findings, we have conducted additional analyses and validation procedures to enhance the robustness of our proposed tumor subtyping method. This involved incorporating more extensive independent samples from a clinical trial and undertaking thorough testing to strengthen the independent prediction power of the identified subtypes.

1. As in most radiomics work, e.g. the seminal Aerts et al. study, the major concern is the independence of the radiomic signature (in this case the habitat-based tumor subtyping) from the simple parameter of tumor volume (Vallières et al. Journal of Nuclear Medicine 2018; 59:640, Welch et al. Radiotherapy and Oncology 2019). Tumor volume is one of the strongest known predictors of disease recurrence and overall survival in lung cancer and it could thus bias the proposed habitat imaging subtypes significance. Extensive safeguards should be applied to rule this out.

Reply: Thank you for pointing out the potential bias introduced by tumor volume in our proposed habitat imaging subtypes and its impact on recurrence-free survival (RFS) and overall survival (OS) prediction in lung cancer. We completely agree in that tumor volume is a well-established predictor of RFS and OS in lung cancer patients. Its potential impact on the significance and interpretation of habitat imaging subtypes is an important consideration. We are committed to ensuring the robustness and accuracy of our results by implementing extensive safeguards to account for this potential bias.

In our revised analysis, we carefully controlled tumor volume as a covariate as well as in subgroup analysis to exam its potential confounding effect on the significance of the proposed habitat imaging subtypes. This approach enables a more accurate evaluation of the independent predictive value of our identified imaging subtypes beyond the potential influence of tumor volume. By taking tumor volume into consideration, a similar performance was observed suggesting the habitat-based tumor subtyping provides additional prognostic information beyond tumor volume.

Additionally, we discussed in the revised manuscript (**Section 2.7, page 14, lines 281-285**) the importance of considering tumor volume as a potential confounding factor, which reads as the following:

“Tumor volume stands out as a pivotal predictor in lung cancer prognosis, exerting a substantial influence on disease recurrence and overall survival³²⁻³⁴. Acknowledging its critical role, we systematically address the potential impact of tumor volume on the significance of our proposed habitat imaging subtypes”.

We have also cited these 3 seminal papers in the revised manuscript.

- Aerts, Hugo JW, et al. "Decoding tumour phenotype by noninvasive imaging using a quantitative radiomics approach." Nature communications 5.1 (2014): 4006.
- Vallières, Martin, et al. "Responsible radiomics research for faster clinical translation." Journal of Nuclear Medicine 59.2 (2018): 189-193.

- Welch, M.L., McIntosh, C., Haibe-Kains, B., Milosevic, M.F., Wee, L., Dekker, A., Huang, S.H., Purdie, T.G., O'Sullivan, B., Aerts, H.J. and Jaffray, D.A., 2019. Vulnerabilities of radiomic signature development: The need for safeguards. *Radiotherapy and Oncology*, 130, pp.2-9.

The measures taken to address this potential bias is discussed in **Section 3.4 (page 18, lines 370-372)** of the revised manuscript. It is also described to address the next question (1.1):

“Univariate and multivariate analysis, after adjustment for tumor volume reveals the independent impact of habitat imaging subtypes in both the discovery and validation cohorts (Supp Table 1a, Supp Table 1b).”

These revisions aim to enhance the validity and interpretation of our results, offering a more comprehensive understanding of the role of habitat imaging subtypes in predicting RFS and OS in our cohorts.

1.1 A multivariable analysis including tumor volume was presented (Table 3), however, only in the integrated cohort (discovery and validation sets combined). Moreover, 12 variables were included which is potentially problematic for generalizability given the number of events. The independent impact of habitat imaging subtypes should be analyzed more in detail as it is the main message of this work. By separate multivariable analyses in the discovery and validation cohorts. Univariable significance and/or exclusion of intercorrelated variables may guide variable selection.

Reply: We thank the reviewer for this constructive suggestion and we further agree on the importance of analyzing the independent impact of habitat imaging subtypes in both the discovery and validation cohorts. In response to your suggestion, we have conducted separate multivariable analyses for these cohorts.

Additionally, we recognize the concern about the number of variables included in our analysis, and we have refined our approach. In the revised version, we have reduced the number of variables, considering univariable significance and excluding intercorrelated variables to enhance the generalizability of our findings (**see updated Table 3 below**). Furthermore, we have carried out additional analysis to compare the habitat imaging subtypes and tumor volume separately in the discovery and validation cohorts (**see newly added Supplementary Table 1 a & b**).

We believe that these updated analyses will strengthen the robustness and clarity of our results. Habitat imaging subtypes were prognostic of RFS and OS irrespective of known risk factors. In multivariate analysis, Habitat imaging subtypes remained prognostic for RFS in discovery and validation cohorts and remained prognostic of OS in the validation cohort. Taken together, these results suggest the additional prognostic value of habitat imaging subtypes beyond known prognostic factors such as tumor volume.

Table 3. Univariate and multivariate Cox regression analysis of recurrence-free survival (RFS) and overall survival (OS) in the integrated study cohort

With the high-risk subtype as the baseline, the low-risk and intermediate-risk imaging subtypes remained independent predictors of RFS (hazard ratio [HR] = 0.38, $P = 0.001^{**}$; HR = 0.52, $P = 0.015^*$, respectively), and the low-risk imaging subtype remained an independent predictor of OS (HR = 0.52, $P = 0.025^*$) in multivariate analysis. * $P < 0.05$; ** $P < 0.01$; *** $P < 0.001$.

Univariate and multivariate analyses of RFS and OS in integrated cohort								
Variable	RFS				OS			
	Univariate		Multivariate		Univariate		Multivariate	
	P value	HR (95% CI)	P value	HR (95% CI)	P value	HR (95% CI)	P value	HR (95% CI)
High - Risk	Reference				Reference			
Intermediate - Risk	0.002**	0.48(0.31-0.76)	0.015*	0.52 (0.31 – 0.88)	0.015*	0.58(0.37-0.90)	0.136	0.68 (0.41 – 1.13)
Low -Risk	<0.001***	0.31(0.20-0.48)	0.001**	0.38 (0.21 – 0.68)	<0.001***	0.43(0.27-0.66)	0.025*	0.52 (0.29 – 0.92)
Age	0.889	1(0.98-1)	---	---	0.009*	1(1-1)	---	---
Male vs Female	0.526	1.1(0.78-1.6)	---	---	0.005**	1.7(1.2-2.5)	---	---
Non-Smoker vs Smoker	0.890	0.97(0.62-1.5)	---	---	0.104	1.5(0.92-2.5)	---	---
Histologic type ^a	0.075	0.69(0.46-1)	0.203	0.75 (0.48 – 1.17)	<0.001***	0.52(0.36-0.76)	0.005**	0.55 (0.36 – 0.83)
Treatment type ^b	<0.001***	2.6(1.8-3.9)	<0.001***	2.07 (1.37 – 3.13)	0.023*	1.6(1.1-2.3)	0.33	1.24 (0.81 – 1.90)
MTV_2.5	<0.001***	1.5(1.2-1.8)	0.801	0.90 (0.39 – 2.09)	0.001**	1.4(1.1-1.7)	0.175	1.89(0.75-4.75)
TLG_2.5	0.001**	1(1-1)	0.657	1.01 (0.96 – 1.06)	0.019*	1(1-1)	0.476	0.97 (0.91 – 1.05)
SUV_max	0.074	1(1-1)	0.396	0.99 (0.97 – 1.01)	0.116	1(1-1)	0.364	0.99 (0.96 – 1.01)
SUV_mean	0.411	1(0.97-1.1)	---	---	0.092	1(0.99-1.1)	---	---
Tumor_vol	<0.001***	1.2(1.1-1.4)	0.973	0.99 (0.70 – 1.41)	0.004**	1.2(1.1-1.3)	0.262	0.81 (0.56 – 1.17)

Supplementary Table 1 (a). Univariate and multivariate Cox regression analysis of recurrence-free survival (RFS) and overall survival (OS) in the discovery cohort. Multivariate analysis included adjustment for tumor volume. Low risk subtype was prognostic of RFS (HR = 0.35, $P < 0.001^{***}$).

Univariate and multivariate analyses of RFS and OS in discovery cohort								
Variable	RFS				OS			
	Univariate		Multivariate		Univariate		Multivariate	
	P value	HR (95% CI)	P value	HR (95% CI)	P value	HR (95% CI)	P value	HR (95% CI)
High - Risk	Reference				Reference			
Intermediate - Risk	0.01*	0.47 (0.26 – 0.84)	0.059	0.54 (0.28 – 1.02)	0.151	0.68 (0.40 – 1.15)	0.374	0.77 (0.43 – 1.4)
Low -Risk	<0.001**	0.30 (0.17 – 0.51)	<0.001***	0.35 (0.19 – 0.65)	0.014*	0.54 (0.33 – 0.88)	0.104	0.62 (0.35 – 1.1)
Tumor_vol	<0.001***	1.3 (1.1 – 1.5)	0.295	1.10 (0.92 – 1.31)	0.021*	1.2 (1 – 1.4)	0.331	1.09 (0.91 – 1.3)

Supplementary Table 1 (b). Univariate and multivariate Cox regression analysis of recurrence-free survival (RFS) and overall survival (OS) in the validation cohort. Multivariate analysis included adjustment for tumor volume. Low risk subtype was prognostic of RFS and OS (HR = 0.15, $P = 0.011^{**}$; HR = 0.11, $P = 0.005^{**}$).

Univariate and multivariate analyses of RFS and OS in validation cohort								
Variable	RFS				OS			
	Univariate		Multivariate		Univariate		Multivariate	
	P value	HR (95% CI)	P value	HR (95% CI)	P value	HR (95% CI)	P value	HR (95% CI)
High - Risk	Reference				Reference			
Intermediate - Risk	0.085	0.49 (0.22 – 1.10)	0.036*	0.26 (0.072 – 0.91)	0.036*	0.4 (0.17 – 0.94)	0.03*	0.24 (0.065 – 0.87)
Low -Risk	0.007**	0.33 (0.14 – 0.74)	0.011**	0.15 (0.033 – 0.64)	<0.001***	0.2 (0.08 – 0.53)	0.005**	0.11(0.023 – 0.51)
Tumor_vol	0.167	1.2 (0.94 – 1.4)	0.247	0.75 (0.461 – 1.22)	0.037*	1.2 (1 – 1.5)	0.344	0.80 (0.504 – 1.27)

1.2 In Supplementary Figures 5-6 Kaplan-Meier curves depict the impact of the habitat imaging subtypes in groups with similar clinicopathological parameters. It is advised to similarly analyze groups formed by tumor volume. Is the habitat imaging subtyping as useful in small as in large tumors?

Reply: We added new supplementary figures during the revision process, and the updated figure numbers are **Supp Fig. 7 and Supp Fig. 8**. As the reviewer suggested, we divided the patients into large vs small tumor groups by taking the median cutoff (59 cc) and plotted the survival in different groups. Kaplan-Meier curves in Supplementary Figures 7 (g) and 8 (g) depict the impact of the High-Risk habitat imaging subtypes in stratifying patients based on tumor volume in predicting RFS ($P < 0.0001$) and OS ($P < 0.0001$). The results demonstrated that habitat imaging subtyping was prognostic in large tumors and as well as in small tumors.

Supplementary Figure 7 (g)

Supplementary Figure 8 (g)

1.3 The correlation of simple tumor-related parameters as tumor volume, MTV, TLG, etc. on the one hand with the habitat imaging subtypes on the other hand should be investigated.

Reply: We have plotted out the correlation matrix between individual features (as stratified into tertiles) and three habitat imaging subtypes. As shown below (**new Supp Table 5**), we observed that 58%, 59%, and 64% correlation between habitat subtypes and stratification by MTV, TLG or tumor volume, respectively; while a sizable of patients are off-diagonal. These results suggest habitat analysis learns novel patterns beyond these conventional CT and PET metrics.

MTV vs Habitat

	MTV_High	MTV_Intermediate	MTV_Low
High risk subtype	80%	16%	5%
Intermediate risk subtype	57%	34%	9%
Low risk subtype	2%	39%	59%

TLG vs Habitat

	TLG_High	TLG_Intermediate	TLG_Low
High risk subtype	80%	13%	8%
Intermediate risk subtype	51%	39%	10%
Low risk subtype	4%	38%	58%

Tumor Volume vs Habitat

	TV_High	TV_Intermediate	TV_Low
High risk subtype	86%	14%	0
Intermediate risk subtype	54%	42%	4%
Low risk subtype	0	36%	64%

1.4 The correlations of tumor volume with the ctDNA data and with the molecular profile findings should be checked to eliminate the risk of these being just a surrogate of tumor volume.

Reply: The reviewer brought up an important point as it has been reported that tumor volume is associated with ctDNA positivity. As suggested, we conducted a correlation analysis between tumor volume and ctDNA molecular profiling data (**see newly added Supp Fig 2**). We observed a significant difference in baseline tumor volume when stratified by baseline ctDNA profiling status ($P = 0.01$). However, no significant difference was observed in baseline tumor volume when stratified by serial ctDNA profiling status ($P = 0.055$). Although there are associations between tumor volume and ctDNA profiling, ctDNA can offer complementary value on top of tumor volume as well as habitat imaging subtypes for predicting patient survival (**see Fig 6c**).

Supp Fig 2 shows the correlations of tumor volume with the ctDNA data.

2. Some additional known risk factors of disease recurrence and/or overall survival should be included in the univariable, multivariable and correlation analyses. For instance, primary tumor location categories (e.g. Tonneau et al. Radiotherapy and Oncology 2023) and lung side/lobe location could potentially be correlated to the habitat imaging subtypes. This potential bias should be excluded.

Reply: Thank you for pointing this out to us. Our radiologist team has added the primary tumor location and lung side/lobe information. We observed that only the right upper lobe is prognostic in our cohorts, which is independent of habitat imaging, see **new Supp Table 2** shown below.

Following sentences were added in the revised manuscript: (**page 19, lines 378-382**)

“When analyzing the primary tumor location, we observed that only the right upper lobe is prognostic in our cohorts that is independent of habitat imaging. High-risk and intermediate-risk imaging subtypes remained prognostic of RFS and OS in the multivariate analysis after adjustment for tumor and lobe location information (Supp Table 2).”

Supplementary Table 2. Univariate and multivariate Cox regression analysis of recurrence-free survival (RFS) and overall survival (OS) in the discovery cohort. Multivariate analysis included adjustment for tumor and lobe location. High-risk and intermediate-risk imaging subtypes were prognostic of RFS (HR = 0.43, P <0.001***; HR = 0.30, P <0.001***) in the multivariate analysis. Both subtypes remained prognostic of OS (HR = 0.45, P = 0.001***; HR = 0.35, P < 0.001***). Only the right upper lobe is prognostic in our cohorts, which is independent of habitat imaging.

Univariate and multivariate analyses of RFS and OS in integrated cohort								
Variable	RFS				OS			
	Univariate		Multivariate		Univariate		Multivariate	
	P value	HR (95% CI)	P value	HR (95% CI)	P value	HR (95% CI)	P value	HR (95% CI)
Imaging Subtypes								
Low – Risk (N = 62)	Reference				Reference			
Intermediate – Risk (N = 164)	<0.001***	0.30 (0.19 – 0.46)	<0.001***	0.30 (0.18 – 0.50)	<0.001***	0.39 (0.25 – 0.60)	<0.001***	0.35 (0.21 – 0.59)
High –Risk (N = 92)	<0.001***	0.43 (0.27 – 0.69)	<0.001***	0.43 (0.26 – 0.71)	0.003**	0.49 (0.31 – 0.78)	0.001**	0.45 (0.27 – 0.73)
Tumor location								
Central (N = 70)	Reference				Reference			
Peripheral (N = 202)	0.131	0.71 (0.45 – 1.1)	0.637	0.89 (0.55 – 1.44)	0.296	0.79 (0.51 – 1.2)	0.861	0.96 (0.60 – 1.54)
Ultracentral (N = 46)	0.583	1.18 (0.66 – 2.1)	0.263	0.71 (0.39 – 1.29)	0.932	0.97 (0.53 – 1.8)	0.109	0.60 (0.32 – 1.12)
Lung side/Lobe location								
Left lower lobe (N = 50)	Reference				Reference			
Left upper lobe (N = 79)	0.907	0.97 (0.55 – 1.70)	0.663	0.88 (0.49 – 1.57)	0.445	0.81 (0.47 – 1.39)	0.337	0.76 (0.44 – 1.33)
Right lower lobe (N = 51)	0.494	0.80 (0.42 – 1.51)	0.239	0.68 (0.36 – 1.29)	0.294	0.72 (0.39 – 1.33)	0.161	0.64 (0.35 – 1.19)
Right middle lobe (N = 22)	0.122	1.74 (0.86 – 3.49)	0.27	1.50 (0.73 – 3.06)	0.692	1.15 (0.57 – 2.32)	0.996	1.00 (0.48 – 2.06)
Right upper lobe (N = 116)	0.028*	0.52 (0.30 – 0.93)	0.032*	0.53 (0.30 – 0.95)	0.012*	0.50 (0.29 – 0.86)	0.015*	0.51 (0.29 – 0.88)

2.1 It is not clear how many patients received postoperative radiotherapy(por). Radiotherapy-related parameters are important risk factors of the studied endpoints: prescription dose, but also tumor location (associated to heart dose exposure) were linked to overall survival after radiotherapy. Radiotherapy-related parameters should thus at least be reported and their correlations with outcome and habitat imaging subtypes investigated.

Reply: Thank you for this constructive suggestion. We observed no correlation between radiotherapy-related parameters and survival after adjustment in multivariate analyses as in the **new Supp Table 3** shown below.

Supplementary Table 3. Univariate and multivariate Cox regression analysis of recurrence-free survival (RFS) and overall survival (OS) in the discovery cohort. Multivariate analysis included adjustment for radiotherapy-related parameters. Intermediate-risk imaging subtypes was prognostic of RFS (HR = 0.36, P = 0.001**) in the multivariate analysis. High and Intermediate – risk subtypes remained prognostic of OS (HR = 0.47, P = 0.009**; HR = 0.28, P < 0.001***). No correlation was observed between radiotherapy - related parameters and survival after adjusting them in multivariate analysis.

Univariate and multivariate analyses of RFS and OS in discovery cohort									
Variable	RFS				OS				
	Univariate		Multivariate		Univariate		Multivariate		
	P value	HR (95% CI)	P value	HR (95% CI)	P value	HR (95% CI)	P value	HR (95% CI)	
Imaging Subtypes									
Low – Risk (N = 40)	Reference				Reference				
Intermediate – Risk (N = 47)	0.004**	0.40 (0.21 – 0.74)	0.001**	0.36 (0.189 – 0.67)	0.001**	0.31 (0.15 – 0.63)	<0.001***	0.28 (0.1396 – 0.57)	
High –Risk (N = 58)	0.069	0.61 (0.35 – 1.04)	0.051	0.58 (0.334 – 1.00)	0.042*	0.57 (0.34 – 0.98)	0.009**	0.47 (0.2714 – 0.83)	
Post-operative Radiotherapy (POR)									
Patients did not receive POR (N = 118)	Reference				Reference				
Patients received POR (N = 27)	0.102	1.6 (0.91 – 2.8)	0.499	0.50 (0.069 – 3.67)	0.284	1.4 (0.75 – 2.7)	0.495	0.30 (0.009 – 9.36)	
Radiotherapy dosage									
No dosage (N = 121)	Reference				Reference				
≤ 54 Gy (N = 17)	0.18	1.6 (0.81 – 3.1)	0.266	3.23 (0.410 – 25.48)	0.519	1.3 (0.58 – 2.9)	0.449	3.86 (0.116 – 127.66)	
> 54 Gy (N = 7)	0.046*	2.5 (1.01 – 6.4)	0.097	6.24 (0.717 – 54.27)	0.051	2.5 (1.00 – 6.4)	0.11	17.58 (0.52 – 589.48)	

3. A comparison of the habitat-based tumor subtype classification with a classical radiomics approach to tumor classification (directly calculating features from the whole tumor volume) could enhance the importance of proposed technique. Does radiomics using habitat imaging perform significantly better?

Reply: Based on your advice, we conducted additional experiments, incorporating classical radiomics features by applying pyradiomics on the entire tumor ROI. However, the classical radiomics model did not achieve robust stratification during validation, as evidenced by the Kaplan-Meier plots below (**new Supp Fig. 4**).

To provide a quantitative assessment of the performance of different prediction models, we introduced the net reclassification improvement (NRI) metric. Our observations revealed that the proposed habitat imaging model outperformed the whole tumor-based radiomics model (see results in following table). The incorporation of the NRI metric strengthens our understanding of the models' comparative performance, emphasizing the superiority of the habitat imaging approach in achieving better predictive accuracy.

Supplementary Fig 4. Comparison of habitat-based tumor subtype classification with classical radiomics approach

Imaging model	NRI (OS)	NRI (RFS)
Classical radiomics approach	Reference	
Habitat Imaging approach	0.216 ($P = 0.04$)	0.257 ($P = 0.03$)

4. The validation dataset was collected from the same institution as a large part of the discovery dataset. No true external validation of the technique was thus performed.

Reply: Thank you for bringing out this point. To address this important question, we performed additional experiments to test our habitat model on a prospective multi-institutional trial – ACRIN 6668/ RTOG 0235 as an external and independent validation cohort. Detailed information can be found at ClinicalTrials.gov with identifier: [NCT00083083](https://clinicaltrials.gov/ct2/show/study/NCT00083083).

In ACRIN 6668, a total of 181 patients were enrolled, and the patient image data were shared by TCIA (<https://wiki.cancerimagingarchive.net/pages/viewpage.action?pageId=39879162>). After image and clinical data collection, 62 were included in our study for quantification using habitat imaging analysis, as detailed in the figure below. The detailed demographic information of this external validation cohort is presented in Table 1.

The habitat imaging subtypes have been externally tested in ACRIN 6668 cohort, where they remained prognostic, as shown in the **new Fig 4e**.

Fig.1b is updated with information about ACRIN dataset as below.

Fig 4e. Imaging subtypes are prognostic for OS in NSCLC patients

Kaplan-Meier curve comparing OS of individuals with low risk (green), high risk (red), and intermediate risk (purple) subtypes with $P = 0.02$ by log-rank test in the external validation cohort.

5. Moreover, no firm prediction model (with Cox model coefficients, MSI feature cut-offs to be used, etc.) has been reported nor validated.

Reply: Thank you for highlighting this aspect. *“In contrast to conventional radiomic studies that extract hundreds of radiomic features with feature selection to build a parsimonious model, we applied unsupervised clustering to*

define imaging subtypes based on the features extracted from tumor subregions. Technically speaking, the clustering approach capturing inner data structure is less prone to overfitting and more robust to small-scale and heterogenous cancer populations, similar to previous studies⁴²⁻⁴⁴. We discussed this important point in the revised manuscript in the DISCUSSION section (page 22, lines 444-450).

Specific comments 1: Line 148: What was the treatment period in the different cohorts?

Reply: “Average treatment period for ICON cohort is 37 days, and average treatment period for PROSPECT cohort is 36.5 days”. We add the information in the revised manuscript (page 16, lines 321-322).

Specific comments 2: Line 212: What was the final pixel resolution? The same resolution for every patient?

Reply: The pixel resolution of patients ranges from 0.5 to 1.5 mm². The images were resized to a final pixel resolution of 1 mm², providing a consistent and standardized basis for further analysis across all patients. We add the information in the revised manuscript (page 11, line 216).

Specific comments 3: Line 221: ‘images were fused’, what does this exactly mean?

Reply: “By image fusion, we meant adding the corresponding pixel values of the different image modalities (CT, CT entropy, PET and PET entropy) to generate texture rich composite image used for the habitat detection”. We clarified this in the revised manuscript (page 12, lines 233-235).

Specific comments 4: Line 237: ‘co-occurring pairs of its neighbors’, please elaborate. Neighbors in 2D or 3D?

Reply: “By co-occurring pairs, we meant that we checked each pixel and find its immediate nearest neighbors in the 3D. Pixels are connected if their edges or corners touch.” We clarified this in the revised manuscript (page 13, lines 251-252).

Specific comments 5: Line 243: How were ‘second-order statistical features’ derived from the MSI matrix?

Reply: “After constructing the MSI matrix based on the pixel connectivity, we consider the MSI matrix as input to the graycoprops function in MATLAB to calculate 4 second-order statistical properties of the MSI matrix including contrast, homogeneity, correlation, and energy.”

Contrast:

Returns a measure of the intensity contrast between a pixel and its neighbor over the whole image.

$\sum_{i,j} |i-j|^2 p(i,j)$. Contrast is 0 for a constant image.

Homogeneity:

Returns a value that measures the closeness of the distribution of elements in the GLCM to the GLCM diagonal.

$\sum_{i,j} p(i,j) / (1 + |i-j|)$. Homogeneity is 1 for a diagonal GLCM.

Correlation:

Returns a measure of how correlated a pixel is to its neighbor over the whole image.

$\sum_{i,j} (i-\mu_i)(j-\mu_j) p(i,j) / \sigma_i \sigma_j$

Correlation is 1 or -1 for a perfectly positively or negatively correlated image. Correlation is NaN for a constant image.

Energy:

Returns the sum of squared elements in the GLCM. $\sum I,jp(I,j)^2$. Energy is 1 for a constant image.

We clarified these definitions in the revised manuscript (page 13, 259-262).

Specific comments 6: Line 260: 'tumor volume', meaning primary gross tumor volume as delineated on CT, in cc?

Reply: Yes.

Specific comments 7: Line 298: What was the median follow-up time and the median OS in the different cohorts? Is median survival time can actually be interpreted as the median follow-up time?

Reply: We have collected the median follow up time and median OS time in different cohorts. The median OS time was defined as the time point at which 50% of the patients have experienced the event. The median OS time is defined as half of the median follow-up time. These data were described in the revised manuscript (Table 1).

Specific comments 8: Line 305: 'CT is high', what does 'high' mean here?

Reply: "For clusters 3 and 4, CT Hounsfield Units (HU) number is high, whereas PET SUV and PET entropy values are low (L). Based on the violin plots in Supp. Fig. 1b, we observed that the CT Hounsfield Units (HU) numbers or the distribution of the CT HU numbers in cluster 3 and 4 is much higher than the distribution of PET SUV values in the same clusters. This indicates a denser tissue". We clarified this in the revised manuscript (pages 16-17, lines 329 - 333).

Specific comments 9: Line 381: 'can be accurately identified', this statement is currently not supported by data. How is 'accurately' defined, and which accuracy metric was calculated?

Reply: We agree with the reviewer and have reworded this sentence as the following:

"So, patients at an elevated risk of recurrence may be identified for potential treatment intensification, closer post-treatment monitoring, and potential enrollment in experimental protocols for more aggressive systemic management." (page 21, lines 434-437)

Specific comments 10: Ref. 27 is not referenced.

Reply: Thank you for the meticulous review. Reference added in the text.

Specific comments 11: Ref. 42 is incomplete.

Reply: Reference added in the manuscript.

"Wu J, Mayer AT, Li R. Integrated imaging and molecular analysis to decipher tumor microenvironment in the era of immunotherapy. *Semin Cancer Biol.* 2022"

Specific comments 12: Table 2: How are the MSI matrix-based features that are referred to as 'interactions between subregions and border' defined? Is the border one of the 8 subregions listed in the MSI matrix?

Reply: Interaction between subregions is captured by looping over all pixels and finding their neighbors. If edges or corners of two pixels (from the same or different subregions) are touching, we consider that to be an interaction and add the value one to the corresponding position in the MSI matrix. The tumor border is also considered in addition to these eight intratumoral subregions.

Specific comments 13: Table 3: Please add the units of the variables in order to interpret HR, e.g. tumor volume in cc?

Reply: The variables have been normalized (z-scored) for the univariate and multi-variate analysis. Thus, we didn't include their units.

Specific comments 14: Figure 5: Tumor volume should be reported instead of tumor diameter size.

Reply: Thank you for the suggestion. Tumor volume is used in the revised manuscript and **Fig. 5 has been updated** as below:

Specific comments 15: Figure 6c: Are the clinicopathological features included in these prediction models? Is the difference between the c-indexes significant? It would additionally be helpful to report the improvement in c-index when adding the habitat imaging risk groups to a baseline clinicopathological feature model (including tumor volume).

Reply: Yes, the clinicopathological features such as age, gender, histology type, metabolic tumor volume and total lesion glycolysis are included in these prediction models. As shown in the figure below, there is improvement in c-index when adding the habitat imaging risk groups to a baseline clinicopathological feature model (including tumor volume), as well as further improvement by including ctDNA data.

We have updated Fig. 6c as following:

Specific comments 16: Suppl. Fig.1: Do the cohorts come independently to the conclusion of splitting into 8 clusters or is there cross-talk between discovery and validation cohorts in the methodology on this point? Line 231 'we further confirmed the consistency of these subregions across the 3 cohorts', how was this achieved?

Reply: There was no crosstalk. We ran these cohorts independently and identified 8 similar clusters in the latent embedding space and then we merged all datasets to create the unified clusters.

Specific comments 17: Altogether, was CT or PET data most informative in the developed habitat image subtype signatures?

Reply: Based on our analysis, the informative values of CT and PET data in the habitat image subtype signatures is likely complementary rather than exclusive as shown in **Supp. Fig 1**. Both modalities contribute unique information about the tumor microenvironment, and their combination enhances the overall characterization of tumors.

Specific comments 18: Suppl. Fig. 4c: No significance result is mentioned for MSI 19 in the discovery cohort.

Reply: The * mark is now **added** to MSI 19 to denote its significance.

Specific comments 19: Suppl. Fig. 7b: What could be the reason for the different behavior observed in the validation cohort for these 3 molecular pathway features?

Reply: We thank the reviewer for this question. We are unsure regarding the root causes of these differences, and it is challenging for us to decipher the causes due to the complex biology. These multiscale radiogenomics correlates could be impacted by many factors, including biological variability, sample heterogeneity, and technical variability. To address these uncertainties, a notable strength of our study lies in the analysis of three independent cohorts with paired PET/CT and RNAseq data. From a statistical perspective, we have employed meta-analysis techniques to enhance the robustness of the radiogenomics analysis. This involves considering putative patterns across different cohorts, providing a comprehensive and more reliable insight into the relationships between radiological and genomic data.

Reviewer #3: Thanks for including me to review this manuscript “Habitat Imaging Subtypes Defined on CT and 18F-FDG PET Complement Circulating Tumor DNA in Predicting Disease Recurrence and Correlate with Distinct Molecular Pathways in Resected Non–Small Cell Lung Cancer”.

Reply: Thank you for your dedication and expertise to review our manuscript. Your feedback and insights have been instrumental in refining and enhancing the quality of our research.

1. This study focused on intratumor heterogeneity by dividing the gross tumor into subregions and analyzing spatial interactions, which were performed in NSCLC patients with early-stage disease. Moreover, the availability of in-house gene expression data, blood-derived ctDNA, and high-quality metadata enabled us to identify the biology behind the high-risk imaging subtype, reinforcing why this subtype was associated with a higher rate of recurrence. Overall, this is a nice study that aimed to build a habitat imaging framework in predicting recurrence-free survival (RFS) following curative surgery. However, there are still some minor concerns about this framework.

Reply: We appreciate your positive comments. Your encouraging remarks provide reassurance that our efforts in presenting the findings in a clear and comprehensible manner have been successful.

2. First, in clinical practice when patients received CT scan and showed ground glass opacity less than 8mm, this subgroup of patients may not receive PET-CT later. They would receive biopsy, surgery or just follow up. So, this framework which was based on both CT and PET might somehow limit the amplification?

Reply: Our study primarily focuses on the surgical population of NSCLC patients, aiming to predict their disease recurrence risk post curative surgery. Recognizing that this demographic typically has access to PET/CT for clinical staging, we understand the challenge of translating our habitat analysis into lung cancer screening settings, especially for patients with small ground-glass opacities (GGOs), where PET/CT may not be readily available. We have acknowledged this limitation in our Discussion (**page 23, lines 481-482**), “*In addition, the habitat imaging requires both PET and CT scans that can limit its clinical application especially in screening setting*”.

3. Furthermore, is there any genomic data that might be associated with predict outcomes such EGFR mutation/ALK fusion?

Reply: In response to your suggestion, we have included the analysis of *EGFR* mutation and *ALK* fusion in our study. We did not observe a correlation between *EGFR* mutations or *ALK* fusions with patient survival, as illustrated in the table below (**New Supp Table 4**). This may be due to the limited sample size of patients with tumors harboring these genomic alterations, especially only 1 patient had *ALK* fusions.

Supplementary Table 4. Univariate and multivariate Cox regression analysis of recurrence-free survival (RFS) and overall survival (OS) in the discovery cohort. Multivariate analysis included adjustment for EGFR mutation and ALK fusion. Intermediate-risk imaging subtypes was prognostic of RFS (HR = 0.35, P = 0.001**) and OS (HR = 0.32, P = 0.001**) in the multivariate analysis. The oncogene mutation status exhibits no correlation with patient survival.

Univariate and multivariate analyses of RFS and OS in discovery cohort									
Variable	RFS					OS			
	Univariate		Multivariate			Univariate		Multivariate	
	P value	HR (95% CI)	P value	HR (95% CI)	P value	HR (95% CI)	P value	HR (95% CI)	
Imaging Subtypes									
Low – Risk (N = 40)	Reference					Reference			
Intermediate – Risk (N = 47)	0.004**	0.40 (0.21 – 0.74)	0.001**	0.35 (0.18 – 0.67)	<0.001***	0.32 (0.16 – 0.62)	0.001**	0.32 (0.162 – 0.65)	
High –Risk (N = 58)	0.069	0.61 (0.35 – 1.04)	0.057	0.59 (0.35 – 1.02)	0.011*	0.49 (0.29 – 0.85)	0.013*	0.50 (0.287 – 0.86)	
EGFR mutation									
Wild Type (N = 120)	Reference					Reference			
Mutant (N = 25)	0.275	1.4 (0.77 – 2.5)	0.097	1.66 (0.91 – 3.03)	0.45	0.75 (0.36 – 1.6)	0.81	0.91 (0.42 – 1.94)	
Anaplastic Lymphoma Kinase (ALK)									
Wild Type (N = 144)	Reference					Reference			
Fusion (N = 1)	0.63	1.6 (0.23 – 12)	0.283	3.05 (0.40 – 23.41)	0.688	0.36 (0.0026 – 50)	0.831	0.57 (0.0031 – 104.05)	

REVIEWERS' COMMENTS

Reviewer #1 (Remarks to the Author):

A very thorough response to review with revisions that clarify all points.

Reviewer #2 (Remarks to the Author):

The authors have significantly improved the manuscript. All comments were answered in detail leading to important newly performed analyses as well as clarifications in the methodological description. The authors should be commended for the extensive additional work performed, especially the external validation dataset they included, where the habitat imaging subtype proved again its significance for overall survival prediction. This way, in my opinion a more robust finding including external validation of the independent prediction power of the habitat imaging subtypes is now presented strengthening the message of this work. I can thus recommend its publication in Nature Communications.

Reviewer #3 (Remarks to the Author):

Thanks for including me again for reviewing this impressive manuscript. This study focused on intratumor heterogeneity by dividing the gross tumor into subregions and analyzing spatial interactions, which were performed in NSCLC patients with early-stage disease. Moreover, the availability of in-house gene expression data, blood-derived ctDNA, and high-quality metadata enabled us to identify the biology behind the high-risk imaging subtype, reinforcing why this subtype was associated with a higher rate of recurrence. Regarding my concerns about the PET-CT scanning, the authors have given the detailed responses. Especially for the genomic data, the researchers have given a further analysis and put it together well in supplementary material. Overall, researchers have given a high-quality response.

RESPONSE TO REVIEWERS' COMMENTS

Reviewer #1: A very thorough response to review with revisions that clarify all points.

Reply: Thank you for your feedback on the revised response. We are glad to hear that the revisions were helpful and that all points were clarified to your satisfaction.

Reviewer #2: The authors have significantly improved the manuscript. All comments were answered in detail leading to important newly performed analyses as well as clarifications in the methodological description. The authors should be commended for the extensive additional work performed, especially the external validation dataset they included, where the habitat imaging subtype proved again its significance for overall survival prediction. This way, in my opinion a more robust finding including external validation of the independent prediction power of the habitat imaging subtypes is now presented strengthening the message of this work. I can thus recommend its publication in Nature Communications.

Reply: We are sincerely grateful for your thorough review and positive feedback on the improvements made to our manuscript. We're delighted that our efforts to address all comments and incorporate additional analyses and clarifications have been recognized. Your recommendation for publication in Nature Communications is truly encouraging and validates our efforts to produce impactful research. We thank you for your support and confidence in our work, and we look forward to contributing to the scientific community through the dissemination of our findings.

Reviewer #3: Thanks for including me again for reviewing this impressive manuscript. This study focused on intratumor heterogeneity by dividing the gross tumor into subregions and analyzing spatial interactions, which were performed in NSCLC patients with early-stage disease. Moreover, the availability of in-house gene expression data, blood-derived ctDNA, and high-quality metadata enabled us to identify the biology behind the high-risk imaging subtype, reinforcing why this subtype was associated with a higher rate of recurrence. Regarding my concerns about the PET-CT scanning, the authors have given the detailed responses. Especially for the genomic data, the researchers have given a further analysis and put it together well in supplementary material. Overall, researchers have given a high-quality response.

Reply: Thank you for your kind words and for taking the time to review our manuscript once again. We greatly appreciate your continued involvement and valuable feedback. We are grateful that our detailed responses addressed your concerns, particularly regarding PET-CT scanning, and that you found the additional analysis of genomic data in the supplementary material to be comprehensive and well-integrated.